# KnowGuard: Knowledge-Driven Abstention for Multi-Round Clinical Reasoning

**Xilin Dang**[1][*] **Kexin Chen**[1,2*]**, Xiaorui Su**[2]**, Ayush Noori**[2]**, Iñaki Arango**[2]**, Lucas Vittor**[2]**,**
**Xinyi Long**[3]**, Yuyang Du**[3]**, Marinka Zitnik**[2][†] **Pheng Ann Heng**[1,4]

[1]Department of Computer Science and Engineering, The Chinese University of Hong Kong
[2]Department of Biomedical Informatics, Harvard Medical School
[3]Department of Information Engineering, The Chinese University of Hong Kong
[4]Institute of Medical Intelligence and XR, The Chinese University of Hong Kong
{xldang23,kxchen}@cse.cuhk.edu.hk, marinka@hms.harvard.edu

## Abstract

In clinical practice, physicians refrain from making decisions when patient information is insufficient. This behavior, known as abstention, is a critical safety mechanism preventing potentially harmful misdiagnoses. Recent investigations have reported the application of large language models (LLMs) in medical scenarios. However, existing LLMs struggle with the abstentions, frequently providing overconfident responses despite incomplete information. This limitation stems from conventional abstention methods relying solely on model self-assessments, which lack systematic strategies to identify knowledge boundaries with external medical evidences. To address this, we propose **KnowGuard**, a novel *investigate-before-abstain* paradigm that integrates systematic knowledge graph exploration for clinical decision-making. Our approach consists of two key stages operating on a shared contextualized evidence pool: 1) an evidence discovery stage that systematically explores the medical knowledge space through graph expansion and direct retrieval, and 2) an evidence evaluation stage that ranks evidence using multiple factors to adapt exploration based on patient context and conversation history. This two-stage approach enables systematic knowledge graph exploration, allowing models to trace structured reasoning paths and recognize insufficient medical evidence. We evaluate our abstention approach using open-ended multi-round clinical benchmarks that mimic realistic diagnostic scenarios, assessing abstention quality through accuracy-efficiency trade-offs beyond existing closed-form evaluations. Experimental evidence clearly demonstrates that KnowGuard outperforms state-of-the-art abstention approaches, improving diagnostic accuracy by 3.93% through effective diagnostic interactions averaging 5.74 conversation turns.

**Code:** https://github.com/IcecreamArtist/KnowGuard

## 1 Introduction

Large language models (LLMs) are designed to generate prompt responses based on given instructions (Brown et al., 2020). However, in clinical decision-making, this tendency becomes problematic, as patient's initial information is often incomplete or ambiguous, requiring iterative, multi-round conversations to be progressively disclosed. In such scenarios, the ability to abstain, i.e., recognizing knowledge boundaries and refraining from answering under uncertainty, is crucial for ensuring the safe and effective deployment of clinical AI systems. Yet, current LLMs struggle with abstention, frequently providing overconfident or premature responses. This behavior prolongs diagnostic interactions, delays decision-making, and increases the cognitive burden on physicians, ultimately undermining trust in AI-assisted workflows (Sun et al., 2025; Kumaran et al., 2025).

Existing abstention methods face two fundamental challenges that limit their suitability for clinical applications. First, LLMs inherently exhibit overconfidence and choice-supportive bias. Tradi-

---

[*]Equal contribution.
[†]Marinka Zitnik is the corresponding author.

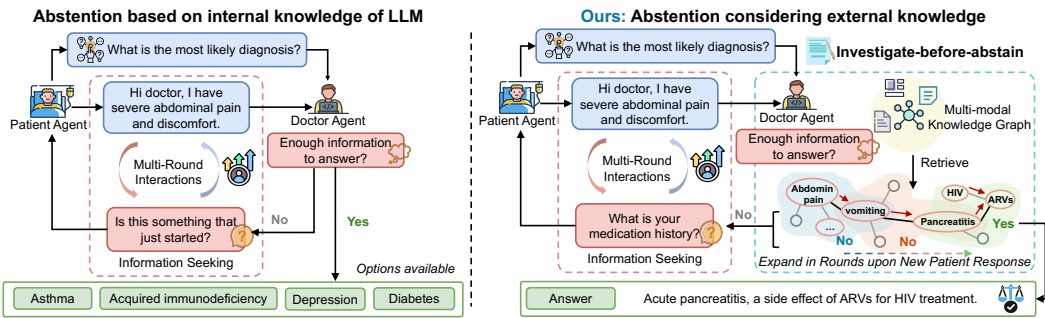

Figure 1: Comparison of abstention approaches in multi-round clinical reasoning. Traditional methods (left) rely on confidence assessment using internal LLM knowledge. Our *investigate-before-abstain* paradigm (right) proactively detects knowledge boundaries through systematic medical knowledge graph exploration, identifying evidence gaps to guide targeted investigation before abstention decisions.

tional confidence-based methods (Tian et al., 2023; Li et al., 2024; Geng et al., 2023) rely on LLM self-assessments to generate confidence scores for abstentions. However, LLMs often inflate their confidence in initial answers, even when faced with contradictory evidence (Tian et al., 2025). This issue could be further exacerbated by the model's reasoning fine-tuning, a post-training method that has been widely applied in recent medical agents (Kirichenko et al., 2025). This overconfidence becomes particularly problematic in multi-round clinical conversations, where models maintain false certainty despite limited patient information. Second, current methods lack robust external knowledge validation methods. Even evidence collection methods, such as the one reported in (Srinivasan et al., 2024), count on internal model knowledge without referencing external medical knowledge. These limitations prove especially concerning in clinical settings, where life-critical decisions require both higher reliability and systematic reasoning grounded in external, verifiable evidence.

**Present work:** This paper incorporates external medical knowledge to address the abstention problem, aiming to ground the LLM's abstention decisions with factual medical evidence beyond its own understanding. The key implementation challenge of the proposed approach is to efficiently and precisely identify the knowledge boundary, i.e., determining whether available evidence is sufficient to support a reliable conclusion. In light of this, a highly structured data representation of the external knowledge source is required to facilitate easier and more accurate boundary identifications. Knowledge graph provides well-organized medical relationships, and is, therefore, a good match to support the systematic reasoning needed for our abstention approach (Gao et al., 2025; Pan et al., 2024).

We highlight that the abstention problem requires a systematic exploration of the medical knowledge graph beyond simple fact retrieval. Under a practical multi-round setup, the system must maintain investigation consistency across interactions and dynamically adapt to new patient information provided. To this end, we propose a novel *investigate-before-abstain* paradigm that grounds abstention decisions in systematic exploration of medical knowledge graphs. This approach progressively investigates knowledge boundaries across rounds, integrating external knowledge with clinical abstention. When new patient details emerge, the system continues exploration rather than restarting, using knowledge conflicts as signals of uncertainty (see Figure 1 for details). Our approach consists of two major stages operating on a shared contextualized evidence pool. The *evidence discovery stage* queries and updates knowledge triplets through graph expansion and direct retrieval based on new patient information. The *evidence evaluation stage* ranks evidence using multiple factors including graph coherence, embedding similarity, LLM selection, temporal decay, and patient population reasoning to identify reliable evidence and facilitate contextualized abstention assessment. Throughout multi-round interactions, this evidence pool functions as a priority queue, continuously updating evidence relevance based on evolving patient context.

In summary, this paper puts forth **KnowGuard**, a multi-round clinical question answering (QA) abstention approach that leverages knowledge graphs with contextualized evidence reasoning. Our

major contributions are summarized as follows: (1) ***Investigate-before-abstain* paradigm**: We replace the unreliable LLM self-assessment scheme with our systematic medical knowledge graph exploration, grounding abstention decisions in factual evidence. (2) **Multi-round knowledge graph reasoning**: We design a two-stage approach with evidence discovery through graph expansion and direct retrieval, followed by evidence evaluation using coherence-aware scoring and demographic-guided reasoning that enables dynamic knowledge expansion adapted to evolving patient information. (3) **Dataset and benchmark**: We establish a new open-ended multi-round clinical benchmark comprising 3,061 cases across three medical datasets. Additionally, we construct a comprehensive medical knowledge graph derived from over 300 WHO guidelines. This knowledge graph encompasses 22k nodes and over 100k edges, integrating multimodal information across text, image, and relation. Unlike existing clinical QA datasets that use multiple-choice formats, our open-ended setting better reflects real clinical conversations and enables proper evaluation of abstention behavior. (4) **Comprehensive system evaluation**: We compare against 5 representative abstention baselines with and without enhancement techniques. Extensive comparisons with state-of-the-art abstention approach show that our method improves diagnostic accuracy by 3.93% with an average of 5.74 effective conversation turns.

## 2 Related Work

**Medical Question Answering Systems.** LLM-powered agents have advanced medical question answering (QA) (Jin et al., 2021; Singhal et al., 2023; Su et al., 2024), which encompasses both multiple-choice and open-ended questions from diverse medical sources. To better reflect real-world clinical practice where physicians often need to gather additional information through iterative questioning, recent research has shifted toward interactive QA frameworks that allow for multi-turn conversations and information seeking (Wang et al., 2025; Johri et al., 2025; Li et al., 2024; Kyung et al., 2025). MediQ (Li et al., 2024) introduced such an interactive QA framework that leverages multi-agent collaboration to encourage agents to abstain from answering when uncertain and actively seek additional information through follow-up questions. However, existing interactive benchmarks predominantly focus on multiple-choice formats, which inadequately reflect real-world clinical scenarios where practitioners typically encounter open-ended questions without predefined answer choices (Nachane et al., 2024). To address this limitation, we develop a multi-round open-ended interactive clinical reasoning benchmark to evaluate free-text responses.

**Abstention Methods.** Effective abstention requires recognizing knowledge boundaries and refraining from answering when evidence is insufficient (Lin et al., 2025; Ni et al., 2025; Kale & Nadadur, 2025). Current approaches include self-assessment methods that rely on internal confidence through uncertainty estimation (Tian et al., 2023), calibration scoring (Geng et al., 2023; Srivastava et al., 2023), and multi-scale rating (Li et al., 2024); consistency-based methods that aggregate multiple model outputs for disagreement detection (Wang et al., 2022); and knowledge-based approaches that incorporate information sources. Long context methods (Tu et al., 2024) retrieve comprehensive medical documents but provide coarse-grained context that fails to pinpoint specific knowledge gaps, leading to information overload rather than targeted evidence discovery. While these methods have shown promise in various domains, they share a fundamental limitation in their reliance on *reactive confidence assessment* rather than *proactive knowledge investigation*. When facing uncertainty, these methods ask "how confident am I?" instead of "what specific evidence am I missing?". KnowGuard introduces the first *investigate-before-abstain* paradigm for multi-round clinical reasoning, which systematically explores knowledge boundaries through targeted evidence discovery guided by medical knowledge graphs.

## 3 Method

### 3.1 Problem Formulation and approach Overview

**Multi-round Abstention Problem Formalization.** We formalize multi-round clinical abstention within an interactive consultation approach that simulates realistic diagnostic scenarios. The Patient Agent maintains complete patient information $\mathcal{K} = \{k_0, k_1, \ldots, k_n\}$ ($n$ pieces in total) and responds truthfully to inquiries by revealing relevant information subsets. The Doctor Agent receives the initial patient presentation $k_0$ and must decide at each round $t$ whether to abstain from diagnosis.

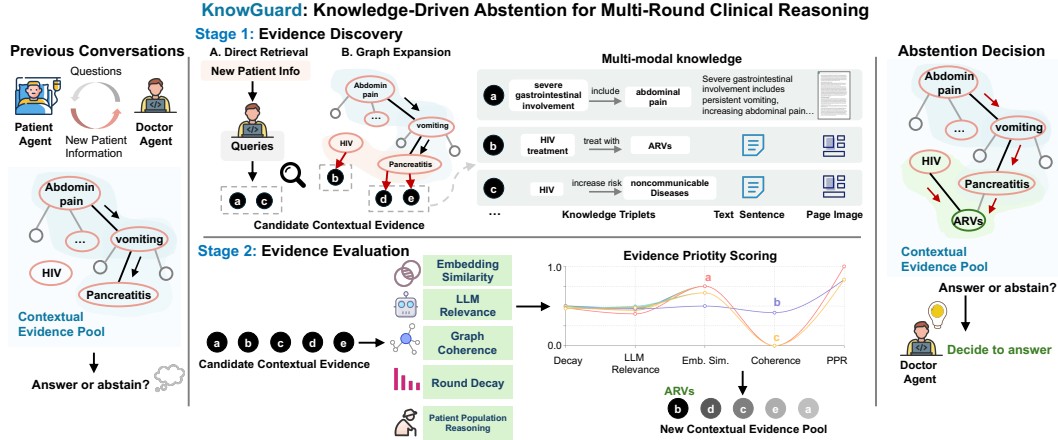

Figure 2: KnowGuard approach for knowledge-driven abstention in clinical reasoning. Our *investigate-before-abstain* paradigm systematically explores medical knowledge graphs to identify evidence gaps before abstention decisions. The Evidence Discovery Stage retrieves multi-modal evidence through dynamic graph expansion and direct retrieval. The Evidence Evaluation Stage adapts exploration priorities through relevance assessment, graph coherence prioritization, demographic weighting, and temporal decay. Final abstention decisions integrate all factors to determine when sufficient evidence exists for diagnosis versus continued investigation.

When abstaining, the agent asks targeted questions $q_t$ to gather additional information; otherwise, it provides a diagnostic answer. At each round $t$, given accumulated patient knowledge $\mathcal{K}_t = \mathcal{K}_{t-1} \cup \{a_t\}$ where $a_t$ represents the patient's response to question $q_t$, the Doctor Agent must make a binary abstention decision:

$$\mathcal{A}_t : \mathcal{K}_t \to \{0, 1\}, \tag{1}$$

where $\mathcal{A}_t = 0$ indicates continued information gathering (abstention) and $\mathcal{A}_t = 1$ indicates sufficient confidence for diagnosis. The core challenge lies in determining the optimal stopping point where $\mathcal{K}_t$ contains sufficient evidence for reliable diagnosis while minimizing unnecessary interaction rounds. Our proposed method focuses on this challenge.

**KnowGuard Approach.** Our *investigate-before-abstain* paradigm replaces unreliable LLM self-assessment with structured medical knowledge exploration. As shown in Figure 2, KnowGuard maintains a contextualized evidence pool $\mathcal{B}_t$ represented as a priority queue of knowledge triplets relevant to the case. The evidence pool evolves cumulatively across conversation rounds, building upon previous discoveries while incorporating new patient information $a_t$. The approach operates through two complementary stages: Evidence Discovery Stage systematically expands $\mathcal{B}_t$ based on patient information, while Evidence Evaluation Stage adapts exploration priorities based on multiple factors, including patient demographics.

**Multi-modal Knowledge Graph.** We construct a comprehensive medical knowledge graph $\mathcal{G} = (\mathcal{V}, \mathcal{E})$ from authoritative medical guidelines, containing $|\mathcal{V}|$ medical entities and $|\mathcal{E}|$ clinical relationships. Each triplet $(h, r, t) \in \mathcal{E}$ is augmented with source text descriptions and document page images, enabling both structured reasoning and contextual validation during evidence discovery. To ensure the knowledge graph remains up-to-date, we have implemented a monthly web scraper module that automatically retrieves newly published WHO guidelines. For more details about the knowledge graph integration, we refer readers to Appendix F.

## 3.2 EVIDENCE DISCOVERY STAGE

**Contextualized Evidence Pool Definition.** The Evidence Discovery Stage operationalizes knowledge boundary exploration through systematic graph investigation. To enable efficient exploration of vast medical knowledge spaces, we maintain a contextualized evidence pool as a priority queue $\mathcal{B}_t = \{(h_i, r_i, t_i, p_i)\}_{i=1}^{|\mathcal{B}_t|}$ of candidate medical triplets (length is $K$), where each triplet $(h_i, r_i, t_i)$

represents a potential reasoning step with priority $p_i$. This bounded representation enables efficient ranking and selection while focusing exploration on the most promising knowledge paths.

**Systematic Evidence Expansion.** The stage performs structured exploration through two complementary retrieval strategies. Graph Expansion-based retrieval identifies triplets connected to entities in current high-priority candidates:

$$\mathcal{T}_{\text{exp}} = \{(h, r, t) \in \mathcal{G} : h \in \mathcal{E}_{\mathcal{B}_t} \text{ or } t \in \mathcal{E}_{\mathcal{B}_t}\}, \tag{2}$$

where $\mathcal{E}_{\mathcal{B}_t}$ represents entities present in current evidence triplet. Direct retrieval first generates queries according to the current patient response $a_t$, and then performs a comprehensive search across the knowledge graph:

$$\mathcal{T}_{\text{query}} = \text{GraphRetrieval}(\mathcal{G}, \text{LLM}_{\text{query}}(a_t)). \tag{3}$$

The retrieved evidence candidates $\mathcal{T}_{\text{candidates}} = \mathcal{T}_{\text{exp}} \cup \mathcal{T}_{\text{query}}$ are fed into the evidence evaluation stage for priority scoring.

### 3.3 EVIDENCE EVALUATION STAGE

The Evidence Evaluation Stage operates on candidate contextual evidence to compute comprehensive priority scores through five complementary factors: Embedding similarity, LLM relevance, graph coherence, round decay, and patient population reasoning.

**Relevance Assessment with Dual Validation.** Each candidate triplet undergoes dual relevance assessment combining embedding similarity (hard relevance) and LLM relevance (soft relevance). Hard relevance measures the semantic similarity between triplet embeddings and the current patient response:

$$s_{\text{sim}}(h, r, t) = \text{cosine}(\text{Embed}(h, r, t), \text{Embed}(a_t)), \tag{4}$$

while soft relevance employs LLM to assess clinical relevance given the current patient context:

$$s_{\text{rel}}(h, r, t) = \text{LLM}_{\text{rel}}(a_t, (h, r, t)). \tag{5}$$

This dual validation ensures both semantic and clinical alignment of evidence investigation.

**Graph Coherence Prioritization.** To maintain reasoning consistency, we prioritize triplets that connect to frequently visited entities, indicating established reasoning pathways:

$$s_{\text{coh}}(h, r, t) = \text{count}_{\mathcal{B}}(h) + \text{count}_{\mathcal{B}}(t), \tag{6}$$

where $\text{count}_{\mathcal{B}}(\cdot)$ tracks cumulative frequency of the entity across all evidence pools throughout the conversation. Higher coherence scores indicate stronger integration with existing paths, enabling systematic knowledge boundary detection rather than random exploration.

**Demographic-guided Priority Weighting.** The stage infers patient demographics and clinical populations from conversation history to prioritize relevant knowledge graph regions. Population inference analyzes accumulated patient information against predefined categories:

$$\mathcal{P}_t = \text{LLM}_{\text{demo}}(\mathcal{K}_t, \mathcal{C}_{\text{pop}}), \tag{7}$$

where $\mathcal{C}_{\text{pop}}$ represents predefined population categories derived from knowledge graph topics, such as adolescents. Triplets belonging to identified patient populations receive enhanced weighting:

$$s_{\text{pop}}(h, r, t) = \begin{cases} \alpha & \text{if } (h, r, t) \in \text{Subgraph}(\mathcal{P}_t) \\ 1 & \text{otherwise,} \end{cases} \tag{8}$$

where $\alpha > 1$ emphasizes population-specific knowledge and $\text{Subgraph}(\mathcal{P}_t)$ denotes triplets relevant to inferred populations.

**Temporal Decay with Round-based Updates.** To balance historical context with current information, the stage applies temporal decay to previously explored knowledge while emphasizing recent evidence. Priority updates follow exponential decay:

$$p_{t+1}(h, r, t) = p_t(h, r, t) \times (1 - w_{\text{decay}}) + p_{\text{new}}(h, r, t) \times w_{\text{decay}}, \tag{9}$$

where $p_{\text{new}}$ reflects priority computed from current round information and $w_{\text{decay}} \in [0, 1]$ controls temporal transition rate.

**Evidence-grounded Abstention Decision.** The final priority combines multiple contextual factors through weighted aggregation:

$$p_{\text{final}}(h, r, t) = (w_{\text{sim}} \cdot s_{\text{sim}} + w_{\text{rel}} \cdot s_{\text{rel}} + w_{\text{coh}} \cdot s_{\text{coh}}) \times s_{\text{pop}}. \tag{10}$$

The contextualized evidence pool maintains top-$K$ triplets: $\mathcal{B}_{t+1} = \text{Top-K}(\mathcal{T}_{\text{candidates}}, p_{\text{final}})$, where $\mathcal{T}_{\text{candidates}} = \mathcal{T}_{\text{exp}} \cup \mathcal{T}_{\text{query}}$. Each triplet is augmented with multi-modal evidence including source text and document images. The final abstention decision integrates structured knowledge evidence with patient context:

$$\mathcal{A}_t = \text{LLM}_{\text{doctor}}(\mathcal{K}_t, \mathcal{B}_t, \{x_{\text{text}}, x_{\text{img}}\}), \tag{11}$$

where the model receives current patient information, top-ranked evidence triplets, and their associated multi-modal content to make informed abstention decisions.

### 3.4 Open-Ended Clinical Reasoning Benchmark

To properly evaluate abstention behavior in realistic clinical scenarios, we establish a multi-round open-ended benchmark that extends beyond existing closed-form evaluations. Traditional multiple-choice formats constrain response options and fail to capture the complexity of real clinical conversations where physicians must formulate comprehensive diagnostic assessments. Following recent advances in automated evaluation (Su et al., 2024), we employ LLM-as-judge methodology to convert closed-ended questions to an open-ended format, enabling more accurate assessment of both diagnostic reasoning quality and abstention appropriateness.

The Judge Agent performs answer matching between free-text predictions and ground truth responses. For originally multiple-choice questions, the judge receives all answer options along with the model's free-text response, without knowing the question content or correct option, and identifies the most semantically similar option:

$$\mathcal{A}_{\text{matched}} = \text{Judge}(\mathcal{A}_{\text{pred}}, \{\text{option}_1, \text{option}_2, \dots, \text{option}_n\}). \tag{12}$$

For originally open-ended questions, the judge performs binary classification to determine whether the prediction aligns with the ground truth answer:

$$\text{Match} = \text{Judge}(\mathcal{A}_{\text{pred}}, \mathcal{A}_{\text{true}}) \in \{\text{Yes}, \text{No}\}, \tag{13}$$

where $\mathcal{A}_{\text{pred}}$ represents the model's free-text response and $\mathcal{A}_{\text{true}}$ denotes the ground truth answer.

## 4 Experiments and Results

We conducted extensive experiments to evaluate the effectiveness of KnowGuard on multi-round clinical abstention, comparing against existing abstention methods on our open-ended interactive clinical reasoning benchmark.

### 4.1 Experimental Settings

**Dataset Construction.** We convert MEDQA (CC-BY-4.0) (Jin et al., 2021), CRAFT-MD (CC-BY-4.0) (Johri et al., 2024), and AFRIMEDQA (CC-BY-NC-SA-4.0) (Nimo et al., 2025) into interactive multi-round formats. Following established protocols (Li et al., 2024), we parse patient records into structured components: age, gender, chief complaint, and additional evidence as atomic facts (Min et al., 2023). Initially, only age, gender, and chief complaint are presented to the Doctor Agent, which must strategically gather missing information through targeted questioning. The resulting interactive datasets are termed ioMEDQA, ioCRAFT-MD, and ioAFRIMEDQA.

**Multi-modal Knowledge Graph Construction.** Our knowledge graph incorporates over 300 WHO guidelines, resulting in 22k medical entities and more than 100k clinical relationships. Each triplet is augmented with source text and document images for comprehensive knowledge boundary detection. Subgraphs are labeled with demographic and disease-specific features extracted from guideline titles and abstracts, enabling patient population reasoning. The system monitors publication dates for automatic updates, ensuring current medical knowledge supports boundary detection decisions.

**Baseline Methods.** We benchmark KnowGuard against representative abstention approaches: Basic (direct question or answer, without explicit abstention step), Binary Decision (Srivastava et al., 2023) (explicit binary abstention), Numerical Score (Tian et al., 2023) (confidence scoring 1-5 with thresholding), Scale Rating (Li et al., 2024) (fine-grained confidence levels with descriptions), and Long Context (Tu et al., 2024) (external document retrieval with full-text processing). We compare the baselines with and without rationale generation (Wei et al., 2022) (generate rationale alongside abstention decision) and self-consistency (Wang et al., 2022) as enhancements. Besides, we compare KnowGuard against calibration-focused methods: Temperature Scaling (Guo et al., 2017) and Conformal Abstention (Yadkori et al., 2024). Since our setting is training-free, we use leave-one-dataset-out partitioning for these methods for fair comparison.

**Metrics and Agent.** We evaluate using Accuracy (ACC), Expected Calibration Error (ECE), Brier Score, and average conversation rounds (avg. Turn) as primary metrics for diagnostic effectiveness and interaction efficiency. All experiments employ GPT-4 (Achiam et al., 2023) as the core agent model, given its widespread adoption and demonstrated capabilities in medical reasoning tasks (Eriksen et al., 2024).

## 4.2 RESULTS

Our *investigate-before-abstain* paradigm achieves the best overall performance, delivering the highest accuracy with the lowest ECE and Brier scores while keeping interaction efficiency competitive (Tables 1 and 2). KnowGuard consistently outperforms all baselines, improving accuracy by 1.07 (5.64%) over the strongest confidence-based methods in basic settings and 1.20 (8.65%) in enhanced settings, and surpassing Long Context by 10.29% (basic) and 14.83% (enhanced) on average. Calibration-based baselines (Temperature Scaling and Conformal Abstention) are overly conservative, often deferring to the maximum round limit (12 turns) with lower accuracy (0.6873/0.7154/0.6916), whereas KnowGuard achieves comparable or better accuracy with far fewer rounds (Table 3). Overall, these results highlight the robustness of systematic knowledge boundary detection over self-assessment-based abstention.

Table 1: Performance comparison of basic methods on open-ended multi-round interactive clinical reasoning, including Basic (implicit), Binary Decision (Srivastava et al., 2023), Numerical Score (Tian et al., 2023), Scale Rating (Li et al., 2024), Long Context (Tu et al., 2024), and KnowGuard (ours).

| Method | ioAFRIMEDQA | | | | ioMEDQA | | | | ioCRAFT-MD | | | |
|---|---|---|---|---|---|---|---|---|---|---|---|---|
| | ACC | Turn | ECE | Brier | ACC | Turn | ECE | Brier | ACC | Turn | ECE | Brier |
| Basic (implicit) | $51.10_{\pm2.40}$ | 8.32 | - | - | $57.83_{\pm2.05}$ | 8.98 | - | - | $54.69_{\pm1.27}$ | 8.31 | - | - |
| Binary Decision | $61.97_{\pm2.83}$ | 8.98 | - | - | $65.95_{\pm1.87}$ | 7.69 | - | - | $64.67_{\pm1.13}$ | 7.85 | - | - |
| Numerical Score | $54.25_{\pm2.69}$ | 1.72 | 0.199 | 0.281 | $61.74_{\pm1.76}$ | 2.51 | 0.123 | 0.246 | $59.35_{\pm1.20}$ | 2.42 | 0.146 | 0.259 |
| Scale Rating | $63.06_{\pm2.34}$ | 5.11 | 0.141 | 0.285 | $64.23_{\pm1.53}$ | 5.15 | 0.135 | 0.260 | $65.40_{\pm1.19}$ | 4.83 | 0.136 | 0.261 |
| Long Context | $57.45_{\pm2.08}$ | 2.01 | 0.283 | 0.323 | $59.95_{\pm1.20}$ | 3.23 | 0.260 | 0.308 | $57.88_{\pm1.58}$ | 3.23 | 0.274 | 0.314 |
| **KnowGuard** | $\mathbf{68.70}_{\pm1.77}$ | 5.26 | **0.050** | **0.236** | $\mathbf{70.98}_{\pm1.98}$ | 5.41 | **0.065** | **0.219** | $\mathbf{66.47}_{\pm1.47}$ | 4.89 | **0.050** | **0.216** |

Table 2: Performance comparison of methods enhanced by Rationale Generation (Wei et al., 2022) and Self-Consistency (Wang et al., 2022) on open-ended multi-round interactive clinical reasoning, including Binary Decision (Srivastava et al., 2023), Numerical Score (Tian et al., 2023), Scale Rating (Li et al., 2024), Long Context (Tu et al., 2024), and KnowGuard (ours).

| Method | ioAFRIMEDQA | | | | ioMEDQA | | | | ioCRAFT-MD | | | |
|---|---|---|---|---|---|---|---|---|---|---|---|---|
| | ACC | Turn | ECE | Brier | ACC | Turn | ECE | Brier | ACC | Turn | ECE | Brier |
| Binary Decision | $64.55_{\pm2.99}$ | 13.82 | - | - | $72.92_{\pm1.47}$ | 13.00 | - | - | $70.01_{\pm1.35}$ | 12.21 | - | - |
| Numerical Score | $58.33_{\pm2.79}$ | 2.63 | 0.190 | 0.272 | $64.23_{\pm1.72}$ | 4.61 | 0.165 | 0.268 | $61.51_{\pm1.17}$ | 4.98 | 0.182 | 0.270 |
| Scale Rating | $61.36_{\pm1.00}$ | 5.31 | 0.122 | 0.286 | $65.52_{\pm1.36}$ | 6.26 | 0.142 | 0.273 | $66.34_{\pm1.89}$ | 5.56 | 0.146 | 0.273 |
| Long Context | $56.80_{\pm0.33}$ | 1.16 | 0.293 | 0.325 | $59.37_{\pm0.84}$ | 3.30 | 0.293 | 0.320 | $58.61_{\pm0.85}$ | 3.29 | 0.293 | 0.329 |
| **KnowGuard** | $\mathbf{73.20}_{\pm1.92}$ | 5.30 | **0.085** | **0.223** | $\mathbf{74.12}_{\pm0.57}$ | 5.40 | **0.099** | **0.203** | $\mathbf{71.96}_{\pm0.98}$ | 6.51 | **0.092** | **0.213** |

Table 3: Comparison of calibration and uncertainty methods on the open-ended reasoning tasks, including Temperature Scaling (Wei et al., 2022) and Conformal Abstention (Yadkori et al., 2024). Carlibration-based methods exhibit overly conservative behavior, consistently deferring to maximum round limit (12 turns) regardless of evidence quality.

| Method | ioAFRIMEDQA | | | | ioMEDQA | | | | ioCRAFT-MD | | | |
|---|---|---|---|---|---|---|---|---|---|---|---|---|
| | ACC | Turns | ECE | Brier | ACC | Turns | ECE | Brier | ACC | Turns | ECE | Brier |
| Temperature Scaling | 0.687 | 12.00 | 0.200 | 0.251 | 0.715 | 12.00 | 0.177 | 0.230 | 0.692 | 12.00 | 0.156 | 0.227 |
| Conformal Abstention | 0.626 | 12.00 | 0.309 | 0.303 | 0.662 | 12.00 | 0.332 | 0.332 | 0.645 | 12.00 | 0.290 | 0.262 |
| **KnowGuard** | **0.732** | 5.30 | **0.085** | **0.223** | **0.741** | 5.40 | **0.099** | **0.203** | **0.720** | 6.51 | **0.092** | **0.213** |

## 5 ANALYSIS

### 5.1 ABLATION STUDIES ON KEY COMPONENTS

To validate the effectiveness of KnowGuard's designs, we conducted systematic ablation studies as shown in Table 4. We progressively evaluate each component's contribution to demonstrate their individual effectiveness. Multi-modal knowledge graph triplets provide substantial improvements over text-only evidence retrieval, demonstrating the value of structured medical knowledge for abstention. The evidence evaluation stage enables systematic exploration by ranking candidate evidence, leading to more targeted abstention decisions. Patient Population Reasoning (PPR) enhances personalized reasoning by considering demographic and disease-specific contexts. Each component contributes meaningfully to both accuracy and efficiency, with the complete system achieving optimal performance across all datasets.

Table 4: Ablation studies of KnowGuard's key designs, including evidence modality of text or multi-modal knowledge graph (KG) triplet, evidence evaluation stage (Evidence Eval.), and patient population reasoning (PPR) factor.

| Component Configuration | | | | ioAFRIMEDQA | | ioMEDQA | | ioCRAFT-MD | |
|---|---|---|---|---|---|---|---|---|---|
| Text evidence | KG evidence | Evidence Eval. | PPR | ACC | avg. Turn | ACC | avg. Turn | ACC | avg. Turn |
| ✓ | ✓ | ✓ | ✓ | 73.20 | 5.30 | 74.12 | 5.40 | 71.96 | 6.51 |
| ✓ | ✓ | ✓ | ✗ | 72.60 | 7.03 | 74.29 | 6.53 | 71.92 | 6.53 |
| ✓ | ✓ | ✗ | ✗ | 66.22 | 2.69 | 70.66 | 3.24 | 68.92 | 3.31 |
| ✓ | ✗ | ✗ | ✗ | 66.02 | 3.33 | 64.79 | 3.25 | 62.73 | 3.30 |
| ✗ | ✗ | ✗ | ✗ | 63.06 | 5.11 | 64.23 | 5.15 | 65.40 | 4.83 |

### 5.2 HYPERPARAMETER STUDIES

Our evidence priority scoring mechanism combines multiple factors for systematic exploration. Table 5 shows sensitivity analysis for each factor. All factors contribute meaningfully to performance, with optimal weights being: embedding similarity $w_{\text{sim}}$ (0.2), LLM relevance $w_{\text{rel}}$ (0.6), graph coherence $w_{\text{coh}}$ (0.35), round decay $w_{\text{decay}}$ (0.5), and patient population reasoning $w_{\text{pop}}$ (1.15). The consistent performance across different weight configurations demonstrates the robustness of our approach, indicating that the method is not overly sensitive to hyperparameter tuning.

Table 5: Sensitivity analysis of evidence evaluation factors. Bold values indicate optimal performance for each factor.

| Dataset | Metric | Embed. Sim. | | | LLM Rel. | | | Graph Coh. | | | Round Decay | | | PPR | | |
|---|---|---|---|---|---|---|---|---|---|---|---|---|---|---|---|---|
| | | 0.1 | 0.2 | 0.3 | 0.5 | 0.6 | 0.7 | 0.25 | 0.35 | 0.45 | 0.4 | 0.5 | 0.6 | 1.10 | 1.15 | 1.20 |
| ioAFRIMEDQA | ACC | 71.41 | **73.20** | 71.99 | 71.02 | **73.20** | 68.51 | 71.41 | **73.20** | 70.44 | 70.25 | **73.20** | 68.32 | 70.25 | **73.20** | 71.22 |
| | Turns | 5.48 | **5.30** | 5.46 | 5.58 | **5.30** | 5.40 | 5.35 | **5.30** | 5.65 | 5.41 | **5.30** | 5.33 | 5.55 | **5.30** | 5.38 |
| ioCRAFT-MD | ACC | 71.10 | **71.96** | 71.67 | 69.58 | **71.96** | 70.22 | 69.50 | **71.96** | 72.47 | 68.78 | **71.96** | 71.51 | 70.30 | **71.96** | 70.24 |
| | Turns | 5.44 | 5.51 | 5.40 | 5.44 | 5.51 | 5.37 | 5.45 | 5.51 | 5.43 | 5.51 | 5.51 | 5.29 | 5.47 | 5.51 | 5.39 |
| ioMEDQA | ACC | 72.77 | **74.12** | 71.24 | 70.59 | **74.12** | 71.64 | 70.27 | **74.12** | 71.72 | 70.84 | **74.12** | 71.16 | 71.64 | **74.12** | 71.98 |
| | Turns | 5.14 | **5.40** | 5.15 | 5.25 | **5.40** | 5.23 | 5.17 | **5.40** | 5.17 | 5.33 | **5.40** | 5.18 | 5.20 | **5.40** | 5.23 |

**Accuracy vs. Conversation Length.** Figure 3(Left) demonstrates the relationships between accuracy and conversation length for our method and traditional self-assessment approaches (Scale Rat-

ing, Numerical Score). KnowGuard shows consistent accuracy improvements with longer conversations, indicating effective knowledge boundary investigation through systematic external knowledge exploration. In contrast, self-assessment methods show steep trajectories where additional rounds provide diminishing returns, reflecting their reliance on internal knowledge. This validates our core hypothesis that proactive knowledge exploration outperforms reactive confidence assessment in multi-round clinical reasoning.

**Confidence Evolution during Conversation.** Figure 3(Middle) shows confidence evolution patterns of our method and Scale Rating throughout conversations. The lengths of different conversations are normalized for intuitive presentation and comparison. Notably, KnowGuard's confidence increases more rapidly than Scale Rating. This indicates that systematic exploration of medical knowledge boundaries enables more targeted information gathering than generic self-assessment.

**Performance on Rare Cases.** Figure 3(Right) compares the accuracy performance on rare diseases. KnowGuard demonstrates substantial advantages over other abstention methods. This suggests that introducing external knowledge as contextual evidence effectively enhances reasoning in challenging cases where traditional self-assessment methods struggle, while the design of patient population reasoning enables targeted exploration of relevant medical subgraphs for more informed abstention decisions.

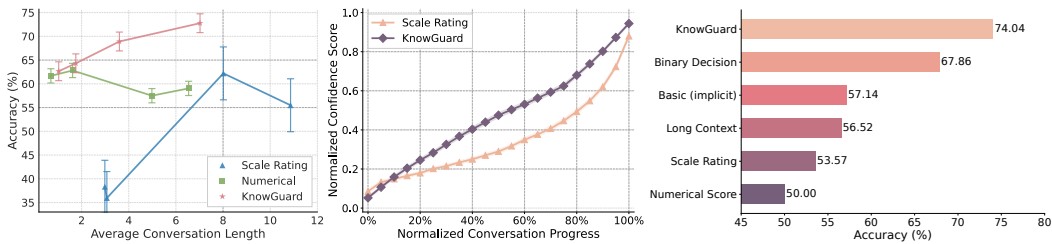

Figure 3: (Left) Systematic evidence exploration enables accuracy improvements with longer conversations, unlike confidence-based self-assessment methods. (Middle) KnowGuard's confidence evolves more rapidly through targeted evidence acquisition compared to generic self-assessment. (Right) *Investigate-before-abstain* paradigm particularly benefits rare disease diagnosis where external knowledge exploration is crucial.

**Case Study.** Figure 4 illustrates an example of KnowGuard's *investigate-before-abstain* paradigm. When presented with abdominal pain symptoms, the system proactively investigates contextual evidence to explore medical knowledge boundaries, ultimately reaching an accurate diagnosis with comprehensive treatment recommendations. This demonstrates how systematic knowledge boundary exploration enables confident decision-making in complex clinical scenarios. See Appendix H and I for more case studies about system robustness with clinical validations.

## 5.3 EFFICIENCY ANALYSIS

Table 6: Comparison of token usage and cost between KnowGuard and Long Context methods.

| Method | Avg Input Tokens/Case | Avg Output Tokens/Case | Avg Total Tokens/Case | Std Dev Total Tokens/Case | Cost/Case |
|---|---|---|---|---|---|
| KnowGuard | 26,319 | 792 | 27,111 | 21,200 | $0.2870 |
| Long Context | 624,307 | 1,361 | 625,668 | 1,286,006 | $6.2714 |
| Overhead | -95.8% | -41.8% | -95.7% | N/A | -95.42% |

As shown in Table 6, KnowGuard achieves a 95.7% reduction in total tokens and 95.4% cost savings per case compared to Long Context, mainly because it uses structured knowledge graph retrieval (targeted triplets with multimodal evidence) instead of full-text document processing. For latency (Table 7), KnowGuard has higher total case latency (+115%) due to more rounds (5.30 vs. 1.16 avg.) to mitigate Long Context's overconfidence. However, per-round latency is 28.5% faster (10.33s vs. 14.44s), which is more clinically relevant in turn-by-turn interactions with human response time

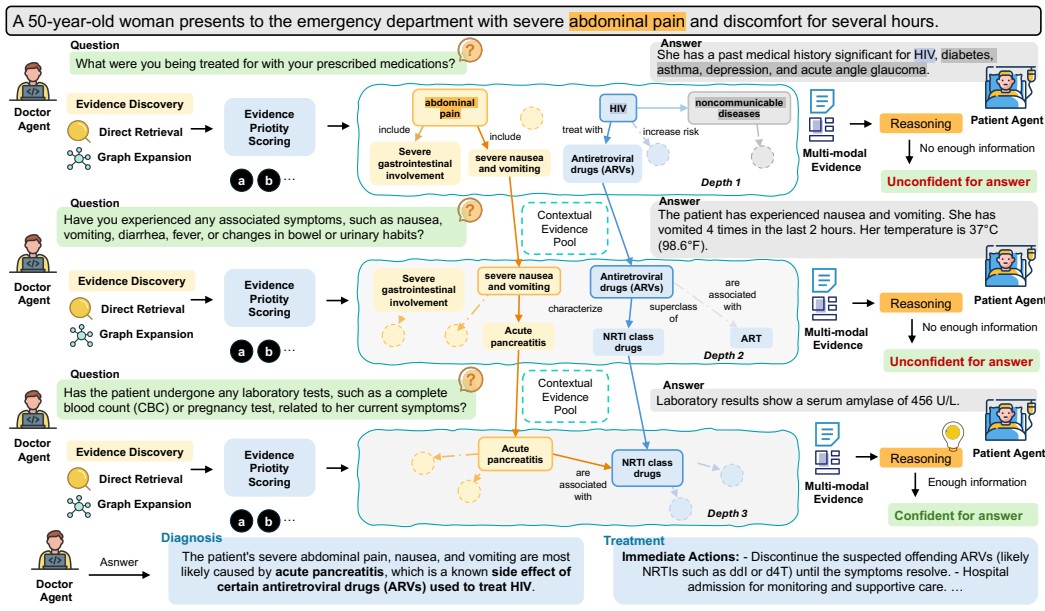

Figure 4: Case study demonstrating KnowGuard's *investigate-before-abstain* paradigm.

Table 7: Comparison of latency between KnowGuard and Long Context methods.

| Method | Avg Latency/Case (seconds) | Avg Latency/Round (seconds) |
|---|---|---|
| KnowGuard | 54.77 | 10.33 |
| Long Context | 25.45 | 14.44 |
| Overhead | +29.32 s (2.15× total time) | -28.46% (1.40× speed up) |

between rounds; thus, KnowGuard provides a 1.4x speedup per interaction and improves responsiveness during active conversations.

# 6 CONCLUSION

In this work, we establish the novel task of open-ended multi-round clinical reasoning and present KnowGuard, an *investigate-before-abstain* paradigm that shifts from internal LLM knowledge to external evidence investigation. Our comprehensive experiments demonstrate that this paradigm shift yields substantial improvements across diverse clinical datasets, with KnowGuard achieving state-of-the-art performance compared to five baseline abstention methods across all benchmarks.

# 7 ETHICS STATEMENT

This work presents a research framework evaluated on academic benchmarks and is not intended for direct clinical use. Real-world deployment requires prospective clinical trials with diverse patient populations, regulatory approval, ongoing bias monitoring (particularly in abstention decisions that could disproportionately affect vulnerable groups), and mandatory oversight by licensed medical professionals. KnowGuard is a research prototype and should not be used for actual medical diagnosis or treatment decisions. All outputs must be reviewed by qualified healthcare providers, and any clinical application must comply with local healthcare regulations and ethical guidelines.

## 8 ACKNOWLEDGEMENT

The work described in this paper was primarily supported by the Research Grants Council of the Hong Kong Special Administrative Region, under Project T45-401/22-N. The authors also acknowledge the partial support from the JC STEM Lab of Advanced Wireless Networks for Mission-Critical Automation and Intelligence, funded by The Hong Kong Jockey Club Charities Trust.

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

## A    STATEMENT ON LLM USAGE

We disclose that LLMs were used solely to aid and polish writing, including covering spell checking, grammar fixes, style refinement, and minor wording suggestions. LLMs did not contribute to any scientific or technical content: all conceptualization, method design, implementation, experiments, result analysis, figures/tables, and conclusions were performed and verified by the authors. All cited works were independently retrieved, fully read, and manually verified using official sources; LLMs were not used to generate or fabricate citations or results.

## B    TAKEAWAYS

The results reveal critical insights for abstention design. First, simply incorporating external knowledge is insufficient for effective abstention, as Long Context's suboptimal performance demonstrates despite accessing comprehensive medical documents. Second, even though LLMs have been trained on extensive medical knowledge, our findings show that introducing external knowledge as contextual evidence at inference time significantly improves abstention decisions and reduces overconfidence in clinical reasoning tasks.

## C    LIMITATIONS AND FUTURE WORK

While KnowGuard demonstrates strong performance on established benchmarks, several limitations warrant attention. The current knowledge graph relies primarily on WHO guidelines and may not capture all clinical knowledge domains. Future work should explore integration with broader medical knowledge sources and real-time knowledge updates. Additionally, the system's performance on highly specialized medical domains requires further evaluation.

## D    ABLATION STUDY ON SCORING COMPONENTS

To validate the necessity of each component in our scoring mechanism, we conducted systematic ablation studies by removing one factor at a time while keeping others active. Table 8 presents the results on ioMEDQA and ioAFRIMEDQA datasets.

| Removed Factor | Components | | | | | Metrics | | Impact |
|---|---|---|---|---|---|---|---|---|
| | Emb. | LLM | Graph | Decay | PPR | Rounds | ACC | Δ ACC |
| *ioMEDQA* | | | | | | | | |
| (None - Full Model) | ✓ | ✓ | ✓ | ✓ | ✓ | 5.40 | 74.12 | baseline |
| Embedding Similarity | × | ✓ | ✓ | ✓ | ✓ | 6.06 | 71.64 | -2.48% |
| LLM Relevance | ✓ | × | ✓ | ✓ | ✓ | 6.26 | 72.68 | -1.44% |
| Graph Coherence | ✓ | ✓ | × | ✓ | ✓ | 5.89 | 70.03 | -4.09% |
| Round Decay (=0) | ✓ | ✓ | ✓ | × | ✓ | 5.87 | 70.62 | -3.50% |
| Round Decay (=1) | ✓ | ✓ | ✓ | × | ✓ | 5.79 | 70.84 | -3.28% |
| PPR | ✓ | ✓ | ✓ | ✓ | × | 6.53 | 74.29 | +0.17% |
| *ioAFRIMEDQA* | | | | | | | | |
| (None - Full Model) | ✓ | ✓ | ✓ | ✓ | ✓ | 5.30 | 73.20 | baseline |
| Embedding Similarity | × | ✓ | ✓ | ✓ | ✓ | 5.27 | 71.02 | -2.18% |
| LLM Relevance | ✓ | × | ✓ | ✓ | ✓ | 5.18 | 70.16 | -3.04% |
| Graph Coherence | ✓ | ✓ | × | ✓ | ✓ | 5.73 | 69.23 | -3.97% |
| Round Decay (=0) | ✓ | ✓ | ✓ | × | ✓ | 5.25 | 71.22 | -1.98% |
| Round Decay (=1) | ✓ | ✓ | ✓ | × | ✓ | 6.53 | 72.21 | -0.99% |
| PPR | ✓ | ✓ | ✓ | ✓ | × | 7.03 | 72.60 | -0.60% |

Table 8: Ablation study results showing the impact of removing individual scoring components. Each row represents a configuration with one component disabled while others remain active.

The ablation study demonstrates that each component serves a distinct and necessary role in our scoring mechanism. **Graph Coherence** emerges as the most critical factor, with its removal caus-

ing the largest accuracy drops (3.97-4.09%), confirming the importance of structured relationship modeling for maintaining knowledge consistency in medical reasoning. **Round Decay** validates the need for adaptive temporal reasoning, as both extreme settings significantly degrade performance: decay=0 (treating all rounds equally) causes 1.98-3.50% drops, while decay=1 (only considering current round) leads to 0.99-3.28% drops. **Embedding Similarity** and **LLM Relevance** prove essential for evidence filtering and semantic matching, with their removal causing 1.44-3.04% accuracy reductions. **PPR** contributes modest but consistent improvements (0.17-0.60%), validating its role in demographic-specific evidence prioritization. These complementary contributions justify our weighted combination approach rather than relying on any single scoring mechanism.

## E  MULTI-ROUND CLINICAL REASONING BENCHMARK

### E.1  DATASET CONSTRUCTION AND OPEN-ENDED CONVERSION

We constructed a comprehensive benchmark for multi-round clinical reasoning by converting traditional closed-form medical datasets to an open-ended format. The benchmark comprises 3,061 cases across three datasets: MEDQA development (1,269 cases), AFRIMEDQA (522 cases), and CRAFT-MD (1,270 cases), as shown in Table 9. We utilized Factscore (Min et al., 2023) to extract atomic facts from patient context following (Li et al., 2024).

AfriMedQA contains both open-ended and multiple-choice questions, which require different evaluation strategies in our judge agent.

Table 9: Dataset composition for multi-round clinical reasoning evaluation.

| Dataset | Size |
|---|---|
| MEDQA development | 1,269 |
| AFRIMEDQA | 522 |
| CRAFT-MD | 1,270 |
| **Total** | **3,061** |

### E.2  PERFORMANCE OF RARE CASES

To evaluate the effectiveness of systematic knowledge boundary detection on challenging diagnostic scenarios, we identify rare disease cases within the benchmark. Using spaCy/scispaCy (Honnibal et al., 2024) and regular expressions, we extract approximately 200 medical terminologies, conduct frequency analysis to select 60 least frequent terms, and utilize LLM validation to identify 25 confirmed rare diseases according to medical consensus (Phan et al., 2024). This analysis demonstrates how knowledge boundary detection addresses insufficient evidence scenarios that traditionally challenge confidence-based abstention methods. The results are shown in Figure 3(Right).

## F  KNOWLEDGE GRAPH INTEGRATION

### F.1  ENTITY/RELATION EXTRACTION

---

**Extraction Prompt**

**Task Instructions**: I am constructing a knowledge graph in the medical field. From this image, please help me extrapolate knowledge such as (`x_name`, `x_type`, `relationship`, `display_relation`, `y_name`, `y_type`, `relevant_description`) in JSON format.

**Input**: Each page in clinical guideline.

**Explanation**:
Please note that the relationship includes but is not limited to ['`protein_protein`', '`drug_protein`', '`contraindication`', '`indication`', '`off-label`

---

```
use', 'drug_drug', 'phenotype_protein', 'phenotype_phenotype',
'disease_phenotype_negative', 'disease_phenotype_positive',
'disease_protein', 'disease_disease', 'drug_effect',
'bioprocess_bioprocess', 'molfunc_molfunc', 'cellcomp_cellcomp',
'molfunc_protein', 'cellcomp_protein', 'bioprocess_protein',
'exposure_protein', 'exposure_disease', 'exposure_exposure',
'exposure_bioprocess', 'exposure_molfunc', 'exposure_cellcomp',
'pathway_pathway', 'pathway_protein', 'anatomy_anatomy',
'anatomy_protein_present', 'anatomy_protein_absent']
```
The display relation includes but is not limited to [`'associated with'`, `'carrier'`, `'contraindication'`, `'enzyme'`, `'expression absent'`, `'expression present'`, `'indication'`, `'interacts with'`, `'linked to'`, `'off-label use'`, `'parent-child'`, `'phenotype absent'`, `'phenotype present'`, `'ppi'`, `'side effect'`, `'synergistic interaction'`, `'target'`, `'transporter'`]
`x_type` and `y_type` include but are not limited to [`'gene/protein'`, `'drug'`, `'effect/phenotype'`, `'disease'`, `'biological_process'`, `'molecular_function'`, `'cellular_component'`, `'exposure'`, `'pathway'`, `'anatomy'`]
`relevant_description` should be a sentence or paragraph extracted from this image, which describes all the relevant information for `x_name` and `y_name`.

**Response Format**: Please provide the information formatted as a JSON object. The structure must strictly adhere to the following requirements:
1. The JSON object should consist exclusively of these keys: "x_name", "x_type", "relationship", "display_relation", "y_name", "y_type" and "relevant_description".
2. The response should be clean and precise: it must not contain ellipses ("..."), backticks ("`"), or any code block identifiers such as "```json". There should be a numerical index for each piece of knowledge.
Please ensure the JSON object is properly formatted with no additional characters or elements outside of the specified structure.

## F.2 THE GRAPH-SEARCH/EXPANSION ALGORITHM

---

**Algorithm 1** KnowGuard: Investigate-Before-Abstain Framework

---

**Require:** Initial patient info $k_0$, Inquiry $I$, Max rounds $R$
**Ensure:** Final answer $\mathcal{A}$ or abstention decision
1: $\mathcal{K}_0 \leftarrow k_0, t \leftarrow 0$
2: $\mathcal{B} \leftarrow \emptyset$ // Initialize contextualized evidence pool
3: **while** $t < R$ **do**
4:   **if** $t = 0$ **then**
5:     $\mathcal{Q} \leftarrow$ EVIDENCEDISCOVERY($\mathcal{K}_t, k_0, I$)
6:   **else**
7:     $\mathcal{Q} \leftarrow$ EVIDENCEDISCOVERY($\mathcal{K}_t, a_t, I$)
8:   **end if**
9:   $\mathcal{B} \leftarrow$ EVIDENCEEVALUATION($\mathcal{Q}, \mathcal{B}, \mathcal{K}_t, t$)
10:   $abstention\_decision \leftarrow$ EVIDENCEGROUNDEDABSTENTION($\mathcal{K}_t, \mathcal{B}, I$)
11:   **if** $abstention\_decision = 1$ **then** // Provide diagnosis
12:     **return** GENERATEANSWER($\mathcal{K}_t, \mathcal{B}, I$)
13:   **else** // Continue investigation
14:     $q_{t+1} \leftarrow$ GENERATEINVESTIGATIVEQUESTION($\mathcal{B}, \mathcal{K}_t$)
15:     $a_{t+1} \leftarrow$ PATIENTRESPONSE($q_{t+1}$)
16:     $\mathcal{K}_{t+1} \leftarrow \mathcal{K}_t \cup a_{t+1}$
17:     $t \leftarrow t + 1$
18:   **end if**
19: **end while**
20: **return** EVIDENCEGROUNDEDABSTENTION($\mathcal{K}_t, \mathcal{B}, I$)

---

---

**Algorithm 2** Evidence Discovery Stage

---

**Require:** Patient info $\mathcal{K}_t$, New patient response $a_t$, Inquiry $I$
**Ensure:** Evidence priority queue $\mathcal{Q}$
1: $queries \leftarrow$ GENERATEEVIDENCEQUERIES($a_t$, $I$)
2: $evidence\_candidates \leftarrow \emptyset$
    // Direct retrieval based on new patient response
3: **for** each $query \in queries$ **do**
4:    $results \leftarrow$ RETRIEVEFROMKG($query$)
5:    $evidence\_candidates \leftarrow evidence\_candidates \cup results$
6: **end for**
    // Expansion-based retrieval from existing evidence pool
7: **if** $\mathcal{B}_{t-1} \neq \emptyset$ **then**
8:    $expansion\_candidates \leftarrow$ EXPANDFROMEXISTINGEVIDENCE($\mathcal{B}_{t-1}$)
9:    $evidence\_candidates \leftarrow evidence\_candidates \cup expansion\_candidates$
10: **end if**
11: $patient\_context \leftarrow$ INFERPATIENTCONTEXT($\mathcal{K}_t$)
12: $\mathcal{Q} \leftarrow \emptyset$
13: **for** each $evidence \in evidence\_candidates$ **do**
14:    $s_{similarity} \leftarrow$ EMBEDDINGSIMILARITY($a_t$, $evidence$)
15:    $s_{relevance} \leftarrow$ LLMRELEVANCE($a_t$, $I$, $evidence$)
16:    $s_{population} \leftarrow$ DEMOGRAPHICWEIGHT($evidence$, $patient\_context$)
17:    $priority \leftarrow (w_{sim} \times s_{similarity} + w_{rel} \times s_{relevance}) \times s_{population}$
18:    ADDTOQUEUE($\mathcal{Q}$, $evidence$, $priority$)
19: **end for**
20: **return** $\mathcal{Q}$

---

**Algorithm 3** Evidence Evaluation Stage

---

**Require:** Current queue $\mathcal{Q}$, New patient response $a_t$, Investigation $q_t$, Round $t$
**Ensure:** Updated evidence pool $\mathcal{Q}'$
1: $new\_context \leftarrow$ FORMINVESTIGATIONCONTEXT($q_t$, $a_t$)
2: $evidence\_queries \leftarrow$ GENERATEEVIDENCEQUERIES($new\_context$, $I$)
    // Reassess existing evidence against new context
3: **for** each $evidence \in \mathcal{Q}$ **do**
4:    $s_{similarity} \leftarrow$ EMBEDDINGSIMILARITY($evidence\_queries$, $evidence$)
5:    $s_{relevance} \leftarrow$ LLMRELEVANCE($new\_context$, $I$, $evidence$)
6:    $s_{coherence} \leftarrow$ GRAPHCOHERENCE($evidence$, visited_evidence)
7:    $p_{new} \leftarrow (w_{sim} \times s_{similarity} + w_{rel} \times s_{relevance} + w_{coh} \times s_{coherence}) \times s_{population}$
8:    $p \leftarrow p \times (1 - w_{decay}) + p_{new} \times w_{decay} \times decay^t$
9:    UPDATEPRIORITY($\mathcal{Q}$, $evidence$, $p$)
10: **end for**
    // Systematic evidence expansion for gap identification
11: $gap\_candidates \leftarrow$ SYSTEMATICEVIDENCEEXPANSION($\mathcal{Q}$, visited_evidence)
12: $gap\_candidates \leftarrow gap\_candidates \cup$ DIRECTEVIDENCERETRIEVAL($evidence\_queries$)
13: **for** each $candidate \in gap\_candidates$ **do**
14:    $gap\_priority \leftarrow$ CALCULATEEVIDENCEPRIORITY($candidate$, $new\_context$, $I$)
15:    ADDTOQUEUE($\mathcal{Q}$, $candidate$, $gap\_priority$)
16: **end for**
17: **return** TOPK($\mathcal{Q}$, $k_{max}$)

---

---

**Algorithm 4** Systematic Evidence Expansion

---

**Require:** Evidence queue $\mathcal{Q}$, Explored evidence $\mathcal{E}_e$
**Ensure:** New evidence candidates $\mathcal{C}$
1: $\mathcal{C} \leftarrow \emptyset$
2: $evidence\_nodes \leftarrow \text{EXTRACTEVIDENCENODES}(\mathcal{Q})$
       // Multi-Hop Expansion for evidence discovery
3: **for** each $node \in evidence\_nodes$ **do**
4:    $related\_evidence \leftarrow \text{GETRELATEDEVIDENCE}(node)$
5:    **for** each $evidence \in related\_evidence$ **do**
6:        $evidence\_id \leftarrow \text{CREATEEVIDENCEID}(node, evidence)$
7:        **if** $evidence\_id \notin \mathcal{E}_e$ **then**
8:            $gap\_potential \leftarrow \text{ASSESSEVIDENCEGAP}(evidence, \mathcal{Q})$
9:            **if** $gap\_potential > \theta_{gap}$ **then**
10:               $\mathcal{C} \leftarrow \mathcal{C} \cup \{evidence\}$
11:               $\mathcal{E}_e \leftarrow \mathcal{E}_e \cup \{evidence\_id\}$
12:            **end if**
13:        **end if**
14:    **end for**
15: **end for**
16: **return** $\mathcal{C}$

---

### F.3 FUSION OF GRAPH SIGNALS AND LLM SCORES

The algorithm for fusion of graph signals and LLM scores are presented in Algorithm 5.

---

**Algorithm 5** Multi-factor Evidence Priority Calculation

---

**Require:** Evidence $evidence$, Clinical context $context$, Inquiry $I$, Round $t$
**Ensure:** Evidence priority score $p$
1: $s_{similarity} \leftarrow \cos(embed(context), embed(evidence))$
2: $s_{relevance} \leftarrow \text{LLMRELEVANCEAGENT}(context, I, evidence)$
3: $s_{coherence} \leftarrow \text{GRAPHCOHERENCESCORE}(evidence, existing\_evidence)$
4: $s_{population} \leftarrow \text{DEMOGRAPHICWEIGHT}(evidence, context)$
       // Multi-factor priority emphasizes evidence gaps
5: $p_{evidence} \leftarrow w_{sim} \times s_{similarity} + w_{rel} \times s_{relevance} + w_{coh} \times s_{coherence}$
6: $p \leftarrow p_{evidence} \times s_{population} \times decay^t$
       // Boost priority for critical evidence gaps
7: **if** $s_{coherence} > \theta_{critical}$ **then**
8:    $p \leftarrow p \times \alpha_{boost}$
9: **end if**
10: **return** $p$

---

### F.4 IMPLEMENTATION DETAILS AND HYPERPARAMETERS

#### F.4.1 EVIDENCE DISCOVERY AND EVIDENCE EVALUATION

We construct a comprehensive medical knowledge graph from current medical guidelines, where each triplet $(h, r, t)$ is augmented with multi-modal evidence, including source text and document images. In the evidence discovery stage, we retrieve triplets with a hard relevance threshold of 0.6 for initial filtering.

The contextual evidence pool maintains $K = 6$ triplets during graph expansion and direct retrieval. The graph expansion is with systematic exploration implemented through beam search (beam size=3, maximum hop depth=2). We utilize OpenAI's text-embedding-ada-002 (Neelakantan et al., 2022) for embedding similarity calculation and FAISS (Douze et al., 2024) as the search engine for efficient knowledge graph storage and retrieval.

### F.4.2 MODEL CONFIGURATION AND EXPERIMENTAL SETUP

We leveraged GPT-4 as the agent backbone with temperature 0.6, top-p 0.9, and maximum of 768 tokens for response generation. Document page images were transferred to base64 format for multimodal input processing. All experiments were conducted three times for stability assessment. For abstention decision-making, we enabled self-consistency checking performed twice with an abstention threshold of 3.5.

Since all methods (Scale Rating, Basic, Binary Decision, Numerical Score, Long Context, and KnowGuard) are training-free approaches, we did not partition the datasets into train/validation/test splits. Instead, all methods were directly evaluated on the complete constructed datasets (ioAFRIMEDQA, ioMEDQA, ioCRAFT-MD), ensuring identical access to dataset information. Our ablation studies, hyperparameter tuning, and baseline threshold adjustments were all performed on the same full datasets, guaranteeing equal evaluation conditions.

### F.4.3 IMPLEMENTATION OF LONG CONTEXT

For the Long Context baseline (Tu et al., 2024), we processed summaries for each medical guideline as query keys. Upon summary selection, the corresponding full medical guideline was provided to the Doctor Agent for abstention decisions.

## G  PROMPTS

### G.1  DIRECT RETRIEVAL QUERY GENERATION PROMPT

---
**Query Generation Prompt**

**Task Instructions**: Based on the following input information, generate 2 optimized search queries to retrieve relevant medical knowledge from a knowledge base.
The queries should:

1. Focus on key symptoms, conditions, or medical concerns

2. Use medical terminology when appropriate

3. Be specific enough to find relevant information

4. Cover different aspects of the patient's condition or question

**Input**:
Generate queries based on patient information to find relevant diagnostic and treatment information.

**Response Format**:

```
Query 1: [your first query]
Query 2: [your second query]
```

**Example**:
*Input*: 45-year-old female with recurrent headaches and nausea
*Output*:

```
Query 1:  migraine headache symptoms nausea photophobia
Query 2:  secondary headache causes women middle-aged
```
---

### G.2  PATIENT POPULATION REASONING PROMPT

---
**Patient Population Reasoning Prompt**

**System Prompt**: *You are a medical expert with extensive experience in clinical diagnosis and treatment.*

**Task Instructions**: Given the patient profile and all conditions, please extract the demographic information and disease information from the patient profile that belong to all conditions. Please
---

ensure the information is accurate. Response with the exact demographic information and disease information, separated by a new line. If there is no demographic information or disease information, please return "None".

**Input Format**: `Patient Profile: [patient description] All Demographics: [demographic categories] All Diseases: [disease categories]`

**Response Format**: `[extracted demographics] [extracted diseases]`
If no information found, return "None" for that category.

**Example 1**:
*Patient Profile*: 35-year-old male with chest pain, shortness of breath, and family history of heart disease. He suffers from high blood pressure
*All Demographics*: Pregnant woman, people with HIV, Adults, Elderly
*All Diseases*: heart disease, diabetes, hypertension, cancer
*Answer*: `Adults heart disease, hypertension`

**Example 2**:
*Patient Profile*: 28-year-old female with fatigue, weight gain, and cold intolerance. She is a smoker.
*All Demographics*: Pregnant woman, people with HIV, Adults, Elderly
*All Diseases*: heart disease, diabetes, hypertension, cancer
*Answer*: `Adults None`

## G.3 LLM RELEVANCE PROMPT

**LLM Relevance Prompt**

**System Prompt**: *You are a medical expert with extensive experience in clinical diagnosis and treatment.*

**Knowledge Relevance Evaluation**: You are evaluating the relevance of medical knowledge to patient cases. Rate each relation's relevance independently on a scale from 0 to 1.
**Scoring Guidelines**:

1. 0.0-0.2: Completely irrelevant or contradictory
2. 0.3-0.4: Low relevance, tangentially related
3. 0.5-0.6: Medium relevance, somewhat helpful
4. 0.7-0.8: High relevance, directly applicable
5. 0.9-1.0: Extremely relevant, critical for diagnosis/treatment

**Input Format**: `Patient Information: [patient details] Current Inquiry: [medical question] Relations to evaluate: [0] [relation1] [1] [relation2] ...`
**Response Format**: `[score1, score2, score3, ...]`
**Example**:
*Input*: 35-year-old male with chest pain, shortness of breath *Relations*: [0] Myocardial infarction causes chest pain [1] Diabetes management with metformin
*Output*: `[0.9, 0.1]`

## G.4 ABSTENTION PROMPT

**Prompt for Knowledge-Driven Abstention**

**System Prompt**: *You are a medical doctor trying to reason through a real-life clinical case. Based on your understanding of basic and clinical science, medical knowledge, and mechanisms underlying health, disease, patient care, and modes of therapy, respond according to the task specified by the user. Base your response on the current and standard practices referenced in medical guidelines.*

**Contextual Evidences**

**Confidence Rating Task**: How confident are you in answering the problem factually using the conversation log and medical knowledge? Choose between the following ratings:

1. **"Very Confident"** - The correct answer is supported by all evidence, and there is enough evidence to eliminate the other possible answers, so the option can be confirmed conclusively.

2. **"Somewhat Confident"** - I have reasonably enough information to tell that the correct answer is more likely than other possible answers.

3. **"Neither Confident or Unconfident"** - There is evident supporting the correct answer, but further evidence is needed to be sure about the answer.

4. **"Somewhat Unconfident"** - There is evidence supporting more than one possible answer; therefore, more questions are needed to further distinguish the answers.

5. **"Very Unconfident"** - There is not enough evidence supporting any answers; the likelihood of giving the correct answer at this point is near random guessing.

**Response Format**:

```
REASON: a one-sentence explanation of why you are or are not
confident and what other information is needed.
DECISION: chosen rating from the above list.
```

## G.5  DECISION MAKING PROMPT

**Decision Making Prompt**

**System Prompt**: *You are a medical doctor trying to reason through a real-life clinical case. Based on your understanding of basic and clinical science, medical knowledge, and mechanisms underlying health, disease, patient care, and modes of therapy, respond according to the task specified by the user. Base your response on the current and standard practices referenced in medical guidelines.*

**Task Instructions**: Assume that you already have enough information from the above question-answer pairs to answer the patient inquiry, use the above information to produce a factual conclusion. Respond with a comprehensive and well-reasoned answer.

## G.6  JUDGE AGENT PROMPT

**Judge Agent Prompt**

**System Prompt**: *You are a medical evaluation expert, tasked with evaluating the match between answers and reference standards.*

**Task 1 - Answer-to-Options Comparison**: Analyze the free answer and determine which multiple choice option is closest to the answer.
**Input Format**: `Free Answer: [generated response] Options: A: [option A] B: [option B] C: [option C] D: [option D] E: [option E] (if applicable)`
**Response Format**: Output only the option letter (A, B, C, D, or E) that has the highest match.

**Task 2 - Yes/No Answer Evaluation**: Analyze the free answer against the ground truth and determine if they are similar.
**Input Format**: `Free Answer: [generated response] Ground Truth Answer: [reference answer]`
**Response Format**: Output only 'A' if the answer is similar to the ground truth answer, 'B' if the answer is completely different. Do not include quotation marks.

**Example 1 - Multiple Choice**: *Free Answer*: The patient likely has pneumonia based on the symptoms of fever, cough, and chest pain.

> *Options*: A: Asthma B: Pneumonia C: Heart failure D: COPD
> *Output*: B
>
> **Example 2 - Yes/No Evaluation**: *Free Answer*: The patient should receive antibiotics and supportive care for pneumonia treatment.
> *Ground Truth*: Antibiotic therapy is recommended for bacterial pneumonia along with supportive measures.
> *Output*: A

## H  CASE STUDY ON SYSTEM ROBUSTNESS

This section presents three detailed case studies that systematically evaluate KnowGuard's robustness under progressively challenging conditions:

- **Incomplete KG**: Testing the system's ability to integrate external evidence with parametric knowledge when diagnostic criteria are partially absent from the knowledge graph.
- **Noisy KG**: Evaluating the filtering mechanism's effectiveness in signal preservation when the knowledge graph contains irrelevant or misleading evidence.
- **Misleading Evidence**: Demonstrating the system's resilience when faced with misleading evidence by actively gathering additional information through multi-round questioning and hypothesis refinement to reach the correct diagnosis.

For each case, we provide comprehensive documentation including the patient presentation, all retrieved evidence with scoring details, knowledge queue evolution across interaction rounds, self-consistency evaluation results, and the final clinical reasoning.

### H.1  CASE STUDY 1: INCOMPLETE KG

**Patient Presentation:** A 28-year-old female reports that, for more days than not over the past 3 years, she has felt "down" and, at times, "mildly depressed." Over this period, she also endorses feeling fatigued, difficulty concentrating, and often sleeping more than in the past.

**Question:** What is the minimum amount of time this patient must exhibit these symptoms in order to meet the diagnostic criteria for dysthymia?

**Retrieved Evidence - Round 0 (Initial)**  All retrieved evidence with their scores before filtering is presented in Table 10.

Table 10: All Retrieved Evidence with Scores Before Filtering - Case 1

| ID | Evidence Content | Embedding Similarity | LLM Score | Coherence Score | Status |
|----|------------------|----------------------|-----------|-----------------|--------|
| 1 | Depressive symptoms (or sub-threshold depression) apply to older adults who have two or more simulta... | 0.4780 | 0.1000 | 0.1000 | Filtered |
| 2 | Early findings report, most common ongoing symptoms (regardless of hospitalization status) are fatig... | 0.4644 | 0.0000 | 0.1000 | Filtered |
| 3 | Depressive symptoms: The presence of distress or some degree of impaired functioning in the absence... | 0.4639 | 0.2000 | 0.1000 | Filtered |

**Table 10 – continued from previous page**

| ID | Evidence Content | Embedding Similarity | LLM Score | Coherence Score | Status |
|----|------------------|----------------------|-----------|-----------------|--------|
| 4 | Fatigue and concentration problems were noted to last beyond 12 weeks in patients with post COVID-19... | 0.4625 | 0.0000 | 0.1000 | Filtered |
| 5 | Mild to moderate depression is characterized by depressive symptoms and some functional impairment;... | 0.5272 | 0.3000 | 0.1000 | Filtered |
| 6 | Persons with chronic HCV infection are more likely to develop cognitive dysfunction, fatigue, and de... | 0.4965 | 0.2000 | 0.1000 | Filtered |
| 7 | Depressive symptoms (or sub-threshold depression) apply to older adults who have two or more simulta... | 0.4826 | 0.1000 | 0.1000 | Filtered |
| 8 | Cognitive deficits may overlap or present in clusters with other neurological and non-neurological d... | 0.4708 | 0.4000 | 0.1000 | Retained |
| 9 | Mild to moderate depression is characterized by depressive symptoms and some functional impairment;... | 0.4837 | 0.4000 | 0.1000 | Retained |
| 10 | Severe depression is characterized by depressive symptoms, functional impairment, agitation or psych... | 0.4646 | 0.3000 | 0.1000 | Retained |
| 11 | If the subjective distress of the patient is in terms of intensity and persistence out of proportion... | 0.4563 | 0.5000 | 0.1000 | Retained |
| 12 | Depressive symptoms can be confused with those of other medical illnesses (i.e. weight loss and fati... | 0.4542 | 0.3000 | 0.1000 | Filtered |
| 13 | Depression is essentially an episodic recurring disorder, each episode lasting usually from a few mo... | 0.5184 | 0.2000 | 0.1000 | Filtered |
| 14 | If the subjective distress of the patient is in terms of intensity and persistence out of proportion... | 0.4647 | 0.5000 | 0.1000 | Retained |
| 15 | Mild to moderate depression is characterized by depressive symptoms and some functional impairment;... | 0.4646 | 0.4000 | 0.1000 | Retained |
| 16 | In about 20% of cases, however, depression follows a chronic course with no remission, especially wh... | 0.4506 | 0.3000 | 0.1000 | Retained |

**Self-Consistency Evaluation** The doctor agent performed abstention assessment through self-consistency evaluation with 2 iterations.

**Iteration 1:**

EXPLANATION: Dysthymia, or persistent depressive disorder, is characterized by depressive symptoms that persist for at least 2 years in adults, as outlined by standard diagnostic criteria (e.g., DSM-5). The patient has reported symptoms lasting 3 years, fulfilling the duration criterion, and the symptoms described align with those of dysthymia. The specific duration criteria are not explic-

itly present in the retrieved evidence, though the patient history provides sufficient information to address the question.

DECISION: Very Confident

**Parsed Score:** 5

**Iteration 2:**

EXPLANATION: The diagnostic criteria for dysthymia (persistent depressive disorder) require symptoms to persist for at least 2 years in adults. The patient has reported symptoms persisting for more than 3 years, which meets the duration criterion. While the retrieved evidence does not include specific diagnostic timeframes, the patient history clearly indicates the duration of symptoms, and no conflicting information is present. The specialty involved is psychiatry or mental health, and sufficient information exists in the patient presentation to answer the question.

DECISION: Very Confident

**Parsed Score:** 5

**Average Confidence Score:** 5.0

**Final Answer** Based on the high confidence score (5.0), the doctor agent proceeded directly to provide the final answer without requesting additional information.

To meet the diagnostic criteria for **dysthymia**, also known as **persistent depressive disorder (PDD)**, a patient must exhibit depressive symptoms for **at least 2 years** in adults. During this period, the symptoms should be present for more days than not and should not remit for longer than 2 months at a time.

**Relevant Diagnostic Criteria:**

1. **Duration**: In adults, the minimum required duration is 2 years. For children and adolescents, this duration is reduced to 1 year.

2. **Symptoms**: Persistent depressive symptoms that are less severe than major depressive disorder but significant enough to cause functional impairment. Symptoms may include:

    - Persistent low mood or feeling "down."
    - Fatigue or low energy.
    - Difficulty concentrating.
    - Changes in sleep patterns (e.g., hypersomnia or insomnia).

**Application to the Patient Case:**

The patient, a 28-year-old female, reports experiencing **low mood, fatigue, difficulty concentrating, and increased sleep** for **more days than not over the past 3 years**. Based on the information provided, she meets the minimum duration criterion for dysthymia and exhibits characteristic symptoms that align with the diagnosis. However, to confirm the diagnosis, further evaluation should exclude other causes (e.g., thyroid dysfunction, anemia, or other medical conditions) and assess the degree of functional impairment caused by these symptoms.

**Next Steps:**

1. **Comprehensive Evaluation**: Perform a full medical and psychiatric history, physical examination, and relevant laboratory tests to rule out organic causes, such as hypothyroidism, anemia, or chronic illness.

2. **Diagnostic Confirmation**: Use standardized screening tools such as the **Patient Health Questionnaire (PHQ-9)** or the **DSM-5 criteria for Persistent Depressive Disorder**.

3. **Management**:

    - **Psychotherapy**: Cognitive-behavioral therapy (CBT) or interpersonal therapy (IPT) are effective first-line treatments.

- **Pharmacotherapy**: Consider antidepressant therapy (e.g., SSRIs or SNRIs) if symptoms are significantly impairing function or if psychotherapy alone is insufficient.
- **Lifestyle Modifications**: Encourage regular physical activity, stress management techniques, and sleep hygiene.

In summary, this patient has exhibited symptoms for longer than the minimum 2-year duration required to meet the diagnostic criteria for dysthymia. A thorough evaluation is necessary to confirm the diagnosis and rule out other conditions, followed by an individualized treatment plan.

**Ground Truth Answer:** Two years.

## H.2 CASE STUDY 2: NOISY KG

**Patient Presentation:** A 56-year-old African American male presents with altered mental status, abdominal pain, and a fever of 100.4F.

**Question:** What is the best treatment for this patient's condition?

**Retrieved Evidence - Round 0 (Initial)** All retrieved evidence with their scores before filtering is presented in Table 11.

Table 11: All Retrieved Evidence with Scores Before Filtering - Round 0, Case 2

| ID | Evidence Content | Embedding Similarity | LLM Score | Coherence Score | Status |
|----|------------------|----------------------|-----------|-----------------|--------|
| 1 | If referral to a facility with diagnostic testing is not feasible, presumptive treatment of severe b... | 0.4990 | 0.9000 | 0.1000 | Retained |
| 2 | For gastrointestinal anthrax, 2 ml of ascitic fluid is collected in a sterile screw-capped container... | 0.4880 | 0.2000 | 0.1000 | Filtered |
| 3 | When empyema is present, fever persists despite antibiotic therapy, and the pleural fluid is cloudy ... | 0.4847 | 0.1000 | 0.1000 | Filtered |
| 4 | We recommend for patients with suspected or confirmed severe COVID-19, the use of empiric antimicrob... | 0.4807 | 0.4000 | 0.1000 | Retained |
| 5 | These guidelines include the management of symptomatic infections related to: lower abdominal pain s... | 0.5519 | 0.3000 | 0.1000 | Filtered |
| 6 | The diagnosis of major infection includes acute pelvic inflammatory disease, characterized by fever ... | 0.5190 | 0.2000 | 0.1000 | Filtered |
| 7 | Guidelines for the management of symptomatic sexually transmitted infections begin with a person pre... | 0.5164 | 0.3000 | 0.1000 | Filtered |
| 8 | There were few missed cases with a syndromic approach to lower abdominal pain, which was heavily val... | 0.5138 | 0.2000 | 0.1000 | Filtered |
| 9 | These guidelines include the management of symptomatic infections related to: lower abdominal pain s... | 0.4531 | 0.7000 | 0.1000 | Retained |

Continued on next page

**Table 11 – continued from previous page**

| ID | Evidence Content | Embedding Similarity | LLM Score | Coherence Score | Status |
|----|------------------|---------------------|-----------|-----------------|--------|
| 10 | Immediate treatment of an acute pelvic inflammatory disease may avert adverse consequences such as c... | 0.4519 | 0.2000 | 0.1000 | Filtered |
| 11 | These guidelines include the management of symptomatic infections related to: lower abdominal pain s... | 0.5513 | 0.7000 | 0.1000 | Retained |
| 12 | Guidelines for the management of symptomatic sexually transmitted infections begin with a person pre... | 0.5191 | 0.1000 | 0.1000 | Filtered |
| 13 | There were few missed cases with a syndromic approach to lower abdominal pain, which was heavily val... | 0.4961 | 0.1000 | 0.1000 | Filtered |
| 14 | Individuals presenting with lower abdominal pain syndrome could suggest the presence of acute pelvic... | 0.4933 | 0.1000 | 0.1000 | Filtered |

**Knowledge Queue After Round 0**  After filtering (LLM score $\geq 0.4$), four knowledge triplets were retained. Table 12 shows the final ranked knowledge after round 0.

Table 12: Knowledge Queue After Round 0 - Case 2

| Rank | Knowledge Content | Priority | PPR Weighted | Round |
|------|-------------------|----------|--------------|-------|
| 1 | If referral to a facility with diagnostic testing is not feasible, presumptive treatment of severe bacterial infections should be considered if the person is seriously ill. | 0.6096 | No | 0 |
| 2 | These guidelines include the management of symptomatic infections related to: lower abdominal pain syndrome. | 0.5997 | Yes | 0 |
| 3 | These guidelines include the management of symptomatic infections related to: lower abdominal pain syndrome. | 0.5313 | Yes | 0 |
| 4 | We recommend for patients with suspected or confirmed severe COVID-19, the use of empiric antimicrobials to treat all likely pathogens, based on clinical judgment, patient host factors and local epidemiology, and this should be done as soon as possible (within 1 hour of initial assessment if possible), ideally with blood cultures obtained first. Antimicrobial therapy should be assessed daily for de-escalation. | 0.3482 | No | 0 |

**Self-Consistency Evaluation**  The doctor agent performed abstention assessment through self-consistency evaluation with 2 iterations.

**Iteration 1:**

REASON: The patient's presentation of altered mental status, fever, and abdominal pain raises significant concern for spontaneous bacterial peritonitis (SBP), particularly in a 56-year-old male who

is likely to have underlying liver disease (e.g., cirrhosis, given the mention of ascitic fluid culture). However, the conversation log does not explicitly confirm the presence of ascites or other key features like a history of liver disease, alcohol use, or laboratory findings (e.g., elevated white blood cell count, low ascitic fluid protein, etc.), which would solidify the diagnosis and guide treatment. While empirical antibiotic therapy (e.g., third-generation cephalosporins like cefotaxime) is standard for suspected SBP, the lack of detailed clinical and laboratory information limits the confidence in confirming the diagnosis.

DECISION: Somewhat Confident

**Parsed Score:** 4

**Iteration 2:**

REASON: Based on the clinical presentation of altered mental status, abdominal pain, and fever in a 56-year-old male, there is a high suspicion for spontaneous bacterial peritonitis (SBP), especially if this patient has underlying cirrhosis and ascites. However, the conversation log does not confirm key features such as the presence of ascites, liver disease, or other risk factors (e.g., alcohol use, hepatitis, or portal hypertension) necessary to confidently diagnose SBP. While empiric antibiotic therapy (e.g., third-generation cephalosporins like cefotaxime) is typically initiated in suspected SBP while awaiting ascitic fluid culture results, the lack of specific details limits definitive confidence in the decision.

DECISION: Neither Confident or Unconfident

**Parsed Score:** 3

**Average Confidence Score:** 3.5

**Follow-up Question Generated**   Based on the initial low confidence score (3.5), the doctor agent generated a follow-up question to gather critical missing information.

**Doctor's Question:** What is your physical examination results, focusing on abdominal findings, and paracentesis results including the serum ascites albumin gradient?

**Patient's Response:** Shifting dullness is noted on physical exam. Paracentesis demonstrates a serum ascites albumin gradient of 1.3 g/dL.

**Knowledge Queue After Round 1 (After Patient Response)**   After receiving the patient's response, the knowledge graph was updated. Three knowledge triplets were replaced with new, more relevant evidence. Table 13 shows the updated knowledge queue.

Table 13: Knowledge Queue After Round 1 - Case 2

| Rank | Knowledge Content | Priority | PPR Weighted | Round |
|---|---|---|---|---|
| 1 | Clinical complications of cirrhosis become manifest, including jaundice, ascites, spontaneous bacterial peritonitis, oesophageal varices and bleeding, hepatic encephalopathy, sepsis and renal failure. | 0.5651 | No | 1 |
| 2 | If referral to a facility with diagnostic testing is not feasible, presumptive treatment of severe bacterial infections should be considered if the person is seriously ill. | 0.5014 | No | 0 |
| 3 | These guidelines include the management of symptomatic infections related to: lower abdominal pain syndrome. | 0.4921 | Yes | 0 |

**Table 13 – continued from previous page**

| Rank | Knowledge Content | Priority | PPR Weighted | Round |
|------|-------------------|----------|--------------|-------|
| 4 | NA therapy can be safely administered even to those with decompensated cirrhosis. | 0.4903 | No | 1 |
| 5 | For women with lower abdominal pain with any of the following conditions, good practice includes referral to surgical or gynaecological assessment: abdominal guarding and/or rebound tenderness. | 0.4000 | No | 1 |
| 6 | We recommend for patients with suspected or confirmed severe COVID-19, the use of empiric antimicrobials to treat all likely pathogens, based on clinical judgment, patient host factors and local epidemiology, and this should be done as soon as possible (within 1 hour of initial assessment if possible), ideally with blood cultures obtained first. Antimicrobial therapy should be assessed daily for de-escalation. | 0.3200 | No | 0 |

**Final Abstention Assessment (With Additional Information)** After receiving the patient's response, the doctor agent performed a second abstention assessment with 2 self-consistency iterations.

**Iteration 1:**

EXPLANATION: The patient is a 56-year-old male with cirrhosis, ascites (confirmed by shifting dullness and paracentesis with a serum ascites albumin gradient of 1.3 g/dL), fever, abdominal pain, and altered mental status. These findings strongly suggest spontaneous bacterial peritonitis (SBP), a common and serious infection in patients with cirrhosis and ascites. According to the retrieved evidence on presumptive treatment of severe bacterial infections, immediate empiric antibiotic therapy is recommended while awaiting ascitic fluid culture results to prevent clinical deterioration. Current guidelines recommend starting third-generation cephalosporins, such as intravenous cefotaxime, as the first-line treatment for suspected SBP.

DECISION: Very Confident

**Parsed Score:** 5

**Iteration 2:**

EXPLANATION: The patient is a 56-year-old African American male with cirrhosis presenting with altered mental status, abdominal pain, fever, and ascites. The serum-ascites albumin gradient (SAAG) of 1.3 g/dL indicates portal hypertension as the cause of ascites, and spontaneous bacterial peritonitis (SBP) is a likely diagnosis given his clinical presentation. SBP is a severe bacterial infection of the ascitic fluid, common in patients with cirrhosis. The retrieved evidence emphasizes that presumptive treatment should be considered for seriously ill patients when diagnostic testing is not immediately available. Empiric antibiotic therapy should be initiated promptly while awaiting ascitic fluid culture results, as delayed treatment can lead to poor outcomes.

DECISION: Very Confident

**Parsed Score:** 5

**Average Confidence Score:** 5.0

**Final Answer** Based on the high confidence score (5.0) after receiving additional patient information, the doctor agent proceeded to provide the final answer.

The patient is a 56-year-old African American male with cirrhosis, ascites, altered mental status, abdominal pain, and fever. The serum ascites albumin gradient (SAAG) of 1.3 g/dL indicates portal hypertension as the likely cause of ascites. The clinical presentation strongly suggests spontaneous bacterial peritonitis (SBP), a common and serious complication of cirrhosis and ascites. SBP is characterized by bacterial infection of ascitic fluid without an obvious intra-abdominal source.

According to guidelines on empiric treatment of severe bacterial infections in seriously ill patients, antibiotic therapy should be initiated promptly before ascitic fluid culture results are available, as delayed treatment increases morbidity and mortality. The recommended empiric treatment consists of third-generation cephalosporins, with cefotaxime (2 g IV every 8 hours) being the preferred antibiotic due to its effectiveness against common SBP pathogens (e.g., Escherichia coli, Klebsiella pneumoniae, and Streptococcus species). Alternatively, ceftriaxone (1-2 g IV daily) may be used.

Patients with SBP and a high risk of renal dysfunction (e.g., serum creatinine ¿1 mg/dL, blood urea nitrogen ¿30 mg/dL, or total bilirubin ¿4 mg/dL) benefit from intravenous albumin to reduce the risk of hepatorenal syndrome. The recommended dosing is 1.5 g/kg of albumin on Day 1 and 1 g/kg of albumin on Day 3.

Additional management includes monitoring for complications such as hepatic encephalopathy, renal dysfunction, and sepsis. Ascitic fluid culture and analysis should be performed to confirm SBP, with an ascitic fluid polymorphonuclear leukocyte (PMN) count $\geq 250$ cells/$\mu$L confirming the diagnosis. Non-selective beta-blockers should be discontinued if the patient is taking them, as they may worsen outcomes in SBP patients with hypotension or renal dysfunction. If SBP is confirmed, prophylactic antibiotics (e.g., norfloxacin or ciprofloxacin) may be considered to prevent recurrence in high-risk patients.

In summary, while awaiting ascitic fluid culture results, the best immediate treatment for this patient is empiric administration of cefotaxime or another third-generation cephalosporin, along with albumin infusion to prevent renal dysfunction. Regular monitoring and supportive care are essential to prevent complications associated with decompensated cirrhosis and SBP.

**Ground Truth Answer:** Cefotaxime.

### H.3 CASE STUDY 3: MISLEADING EVIDENCE

**Patient Presentation:** A 4-year-old boy is brought to the emergency department for evaluation after falling.

**Question:** What is the underlying mechanism of this patient's falls?

**Retrieved Evidence - Round 0 (Initial)** The system initially retrieved 6 pieces of medical knowledge evidence. Table 14 shows all retrieved evidence with their scores before and after ranking.

Table 14: All Retrieved Evidence with Scores - Round 0, Case 3

| ID | Evidence Content | Embedding Similarity | LLM Score | Coherence Score | Status |
|----|------------------|----------------------|-----------|-----------------|--------|
| 1 | Medical conditions that increase the risk of falls include: musculoskeletal disease. | 0.5896 | 0.6000 | 0.1000 | Retained |
| 2 | Medical conditions that increase the risk of falls include: low systolic blood pressure. | 0.4843 | 0.5000 | 0.1000 | Retained |
| 3 | Medical conditions that increase the risk of falls include: visual impairment. | 0.5084 | 0.7000 | 0.1000 | Retained |
| 4 | History of head injury is considered in cases of child presenting with lethargy, unconsciousness or convulsions. | 0.5127 | 0.8000 | 0.1000 | Retained |

Continued on next page

**Table 14 – continued from previous page**

| ID | Evidence Content | Embedding Similarity | LLM Score | Coherence Score | Status |
|----|------------------|---------------------|-----------|-----------------|--------|
| 5 | Examination of signs of head trauma or other injuries is considered in cases of child presenting with lethargy, unconsciousness or convulsions. | 0.5695 | 0.5000 | 0.1000 | Retained |
| 6 | On examination, look for altered level of consciousness, neck stiffness, repeated convulsions, bulging fontanelle in infants, non-blanching petechial rash or purpura, lethargy, irritability, and evidence of head trauma suggesting possible recent skull fracture. | 0.5430 | 0.7000 | 0.1000 | Retained |

**Knowledge Queue After Round 0**  Table 15 shows the ranked knowledge queue after Round 0.

Table 15: Knowledge Queue After Round 0 - Case 3

| Rank | Knowledge Content | Priority | PPR Weighted | Round |
|------|-------------------|----------|--------------|-------|
| 1 | Medical conditions that increase the risk of falls include: musculoskeletal disease. | 0.6916 | No | 0 |
| 2 | Medical conditions that increase the risk of falls include: low systolic blood pressure. | 0.5934 | No | 0 |
| 3 | Medical conditions that increase the risk of falls include: visual impairment. | 0.5915 | No | 0 |
| 4 | History of head injury is considered in cases of child presenting with lethargy, unconsciousness or convulsions. | 0.5594 | No | 0 |
| 5 | Examination of signs of head trauma or other injuries is considered in cases of child presenting with lethargy, unconsciousness or convulsions. | 0.5243 | No | 0 |
| 6 | On examination, look for altered level of consciousness, neck stiffness, repeated convulsions, bulging fontanelle in infants, non-blanching petechial rash or purpura, lethargy, irritability, and evidence of head trauma suggesting possible recent skull fracture. | 0.5050 | No | 0 |

**Self-Consistency Evaluation - Round 0**  The doctor agent performed abstention assessment with 2 self-consistency iterations.

**Iteration 1:**

EXPLANATION: The conversation log provides very limited information about the patient's history, circumstances of the fall, symptoms, or any contributing factors. There is no information about the patient's neurological status, musculoskeletal abnormalities, visual impairment, cardiovascular symptoms, or any other relevant medical conditions that might explain the fall. Without further details, it is impossible to determine the underlying mechanism of the patient's falls.

DECISION: Very Unconfident

**Parsed Score:** 1

**Iteration 2:**

EXPLANATION: The patient is a 4-year-old boy who has fallen. The conversation log does not provide sufficient information regarding the nature of the fall, the patient's medical history, developmental milestones, or any symptoms such as altered consciousness, convulsions, or physical examination findings. Retrieved evidence suggests various potential causes for falls in children, including musculoskeletal disease, visual impairment, low systolic blood pressure, and head trauma, but none of these can be confirmed or ruled out based on the current information.

DECISION: Very Unconfident

**Parsed Score:** 1

**Average Confidence Score:** 1.0

**Follow-up Question Generated - Round 1**    Based on the very low confidence score (1.0), the doctor agent generated a follow-up question.

**Doctor's Question:** Did the child lose consciousness before, during, or after the fall?

**Patient's Response:** The patient cannot answer this question, please do not ask this question again.

**Knowledge Queue After Round 1**    After the first interaction, the knowledge graph was updated. Table 16 shows the updated knowledge queue.

Table 16: Knowledge Queue After Round 1 - Case 3

| Rank | Knowledge Content | Priority | PPR Weighted | Round |
|------|-------------------|----------|--------------|-------|
| 1 | Medical conditions that increase the risk of falls include: musculoskeletal disease. | 0.6211 | No | 0 |
| 2 | Medical conditions that increase the risk of falls include: visual impairment. | 0.5501 | No | 0 |
| 3 | Medical conditions that increase the risk of falls include: low systolic blood pressure. | 0.5345 | No | 0 |
| 4 | Progressive neurological impairment, microcephaly, delay in achieving developmental milestones, hypertonia or mental confusion. | 0.5258 | No | 1 |
| 5 | Neurodevelopment (development score at 18 months) in VLBW infants. | 0.5250 | No | 1 |
| 6 | History of head injury is considered in cases of child presenting with lethargy, unconsciousness or convulsions. | 0.5142 | No | 0 |

**Self-Consistency Evaluation - Round 1    Iteration 1:**

EXPLANATION: The patient is a 4-year-old boy with a history of delayed developmental milestones (walking at 18 months), which suggests possible neurodevelopmental delay, a known risk factor for falls. However, without additional history or physical examination findings such as visual impairment, musculoskeletal abnormalities, or neurological symptoms, the exact underlying mechanism remains unclear. Further information like imaging studies, physical examination findings (e.g., gait abnormalities, muscle tone), or family history would be needed for a more definitive answer.

DECISION: Neither Confident or Unconfident

**Parsed Score:** 3

**Iteration 2:**

EXPLANATION: The patient is a 4-year-old boy with delayed motor development (walking at 18 months). Delayed developmental milestones are associated with neurological impairment and can increase the risk of falls. However, the exact underlying mechanism (e.g., cerebral palsy, muscular dystrophy, metabolic disorder, or other neurodevelopmental conditions) cannot be determined without further clinical information, physical examination findings, or diagnostic testing.

DECISION: Neither Confident or Unconfident

**Parsed Score:** 3

**Average Confidence Score:** 3.0

**Follow-up Question Generated - Round 2   Doctor's Question:** Does the child have any history of neurological conditions, such as seizures or developmental delays?

**Patient's Response:** His parents report that he did not walk until he was 18 months old.

**Self-Consistency Evaluation - Round 2   Iteration 1:**

EXPLANATION: Based on the patient's history of developmental delay (walking at 18 months) and recurrent falls, the most likely underlying mechanism for this patient's fall is neurological dysfunction or developmental delay. However, without additional clinical details, such as physical examination, imaging studies, or laboratory tests, the exact mechanism cannot be conclusively determined.

DECISION: Neither Confident or Unconfident

**Parsed Score:** 3

**Iteration 2:**

EXPLANATION: The patient's history of delayed motor milestones (walking at 18 months) and recurrent falls suggests a neurodevelopmental disorder or neuromuscular condition. However, without additional information such as physical examination findings (e.g., muscle tone, reflexes, gait analysis), imaging, or laboratory results, the specific underlying mechanism cannot be determined with certainty.

DECISION: Neither Confident or Unconfident

**Parsed Score:** 3

**Average Confidence Score:** 3.0

**Follow-up Question Generated - Round 3   Doctor's Question:** Has the child experienced any previous falls or frequent stumbling episodes?

**Patient's Response:** He has fallen multiple times in the last year.

**Self-Consistency Evaluation - Round 3**

**Knowledge Queue After Round 3**   Table 17 shows the updated knowledge queue after Round 3.

Table 17: Knowledge Queue After Round 3 - Case 3

| Rank | Knowledge Content | Priority | PPR Weighted | Round |
|------|-------------------|----------|--------------|-------|
| 1 | Medical conditions that increase the risk of falls include: musculoskeletal disease. | 0.6141 | No | 0 |
| 2 | Medical conditions that increase the risk of falls include: visual impairment. | 0.5722 | No | 0 |

**Table 17 – continued from previous page**

| Rank | Knowledge Content | Priority | PPR Weighted | Round |
|------|-------------------|----------|--------------|-------|
| 3 | Progressive neurological impairment, microcephaly, delay in achieving developmental milestones, hypertonia or mental confusion. | 0.5355 | No | 1 |
| 4 | Medical conditions that increase the risk of falls include: low systolic blood pressure. | 0.5126 | No | 0 |
| 5 | Neurodevelopment (development score at 18 months) in VLBW infants. | 0.4724 | No | 1 |
| 6 | Developmental and neurological assessment should be performed with the full engagement of caregivers to identify developmental delays and other nervous system abnormalities such as epilepsy and disorders of movement, posture and swallowing. | 0.4664 | No | 2 |

**Iteration 1:**

EXPLANATION: The patient demonstrates developmental delay (walking at 18 months) and a history of frequent falls, which could indicate underlying neurological dysfunction or musculoskeletal issues. Additional assessments, such as neurological examination, musculoskeletal evaluation, and possibly imaging studies, are necessary to confirm the cause.

DECISION: Neither Confident or Unconfident

**Parsed Score:** 3

**Iteration 2:**

EXPLANATION: The patient's delayed motor development and recurrent falls suggest possible neurological or neuromuscular pathology. However, definitive diagnosis requires further clinical evaluation including physical examination, developmental assessment, and potentially imaging or genetic testing.

DECISION: Neither Confident or Unconfident

**Parsed Score:** 3

**Average Confidence Score:** 3.0

**Follow-up Question Generated - Round 4   Doctor's Question:** Has the child undergone any recent neurological or developmental assessments, and if so, what were the findings?

**Patient's Response:** The patient cannot answer this question, please do not ask this question again.

**Follow-up Question Generated - Round 5   Doctor's Question:** Has the child experienced any difficulty with coordination, balance, or muscle weakness?

**Patient's Response:** He has fallen multiple times in the last year. When standing up, the patient uses his hands against his knees and thighs to slowly push himself up into a standing position.

**Knowledge Queue After Round 5 (Final)**   Table 18 shows the final knowledge queue after all interactions.

Table 18: Final Knowledge Queue After All Interactions - Case 3

| Rank | Knowledge Content | Priority | Subgraph | Round |
|------|-------------------|----------|----------|-------|
| 1 | Medical conditions that increase the risk of falls include: musculoskeletal disease. | 0.6140 | No | 0 |
| 2 | Medical conditions that increase the risk of falls include: visual impairment. | 0.5661 | No | 0 |
| 3 | Perform symptom directed physical examination during routine follow-up visit. | 0.5350 | No | 5 |
| 4 | Progressive neurological impairment, microcephaly, delay in achieving developmental milestones, hypertonia or mental confusion. | 0.5306 | No | 1 |
| 5 | Additional neurological manifestations reported include weakness. | 0.5109 | No | 5 |
| 6 | Medical conditions that increase the risk of falls include: low systolic blood pressure. | 0.4994 | No | 0 |

**Self-Consistency Evaluation - Final (Round 5)**    After receiving information about Gower's sign, the confidence level increased significantly.

**Iteration 1:**

EXPLANATION: Based on the information provided, the 4-year-old boy demonstrates signs of proximal muscle weakness, as evidenced by the use of his hands to push himself up from a seated position (Gowers' sign). This, coupled with delayed developmental milestones (e.g., walking at 18 months) and frequent falls, strongly suggests an underlying neuromuscular disorder, such as Duchenne Muscular Dystrophy (DMD).

DECISION: Very Confident

**Parsed Score:** 5

**Iteration 2:**

EXPLANATION: The patient exhibits classic signs of proximal muscle weakness including Gower's sign (using hands to push up from sitting), delayed motor milestones (walking at 18 months), and recurrent falls. These findings are highly suggestive of Duchenne muscular dystrophy, a progressive neuromuscular disorder caused by dystrophin deficiency.

DECISION: Very Confident

**Parsed Score:** 5

**Average Confidence Score:** 5.0

**Final Answer**    Based on the high confidence score (5.0) after multiple rounds of information gathering, the doctor agent provided the final answer.

The most likely underlying mechanism of this 4-year-old boy's falls is proximal muscle weakness, potentially due to a neuromuscular disorder such as Duchenne muscular dystrophy (DMD). Key findings include: (1) Developmental delay: The child did not walk until 18 months, which is later than the typical developmental milestone. (2) Frequent falls: Recurrent falls over the past year suggest impaired coordination, muscle weakness, or balance issues. (3) Gower's sign: The child uses his hands against his knees and thighs to push himself up into a standing position, which is characteristic of proximal muscle weakness and commonly seen in Duchenne muscular dystrophy. Musculoskeletal disease is a recognized risk factor for falls, and neurological dysfunction with progressive impairment, delay in achieving developmental milestones, and muscle weakness are hallmark features of neuromuscular disorders like DMD. While visual impairment is also a recognized cause of falls

in children, the patient has no visual complaints and the presence of Gower's sign is highly specific for proximal muscle weakness consistent with DMD. While other conditions such as cerebral palsy, spinal muscular atrophy, or metabolic myopathies could contribute to similar symptoms, the combination of delayed walking, recurrent falls, and Gower's sign strongly favors Duchenne muscular dystrophy. Prompt evaluation and diagnosis are essential for initiating appropriate management.

**Ground Truth Answer:** Duchenne Muscular Dystrophy.

## H.4 ROBUSTNESS CASE ANALYSIS

**Case study 1: incomplete KG** This case demonstrates KnowGuard's ability to handle incomplete knowledge graphs by leveraging the agent's parametric knowledge. From the initial 16 retrieved triplets, the system filtered out 8 low-relevance items (LLM score $\leq 0.4$) while retaining 8 contextually appropriate pieces of evidence covering depression severity classification, symptom duration, and diagnostic considerations. Critically, none of the retrieved evidence explicitly contained the "2-year duration criterion" for dysthymia—the essential answer to the question. Despite this knowledge gap in the external graph, the priority-ranked knowledge queue (Table 10) shows appropriate focus on diagnostic thresholds and symptom persistence (e.g., "If the subjective distress of the patient is in terms of intensity ..." has LLM relevance of 0.5). These contextually relevant but incomplete triplets provided sufficient framework for the doctor agent to activate its internal medical knowledge and correctly identify the 2-year requirement through parametric reasoning. The self-consistency evaluation yielded a perfect confidence score of 5.0 across both iterations, reflecting high certainty in the synthesized answer. This case establishes a key capability: when the knowledge graph lacks the precise answer but provides relevant diagnostic context, KnowGuard successfully bridges the gap by integrating external contextual cues with the agent's parametric medical expertise, producing accurate responses without requiring additional exploration rounds.

**Case study 2: noisy KG** This case demonstrates KnowGuard's filtering effectiveness when the knowledge graph contains substantial noise. In Round 0, the initial retrieval produced 14 triplets with significant contamination: 10 pieces (71.4%) focused on unrelated conditions such as pelvic inflammatory disease, sexually transmitted infections, and COVID-19 management—topics semantically similar to "abdominal pain" and "fever" but clinically irrelevant to the patient's actual presentation of spontaneous bacterial peritonitis (SBP) in cirrhosis. The multi-stage filtering mechanism successfully identified and removed these misleading candidates: LLM-based relevance scoring assigned low scores (0.1-0.3) to noise while recognizing genuinely useful evidence such as "presumptive treatment of severe bacterial infections" (LLM score 0.9). Crucially, after the follow-up question elicited critical information about cirrhosis and ascites, Round 1 retrieval automatically replaced three low-priority noisy triplets with highly specific evidence (e.g., "clinical complications of cirrhosis...spontaneous bacterial peritonitis" ranked 1st with priority 0.5651), demonstrating adaptive signal preservation. The confidence progression—from 4.0 (somewhat confident) to 5.0 (very confident)—occurred not through overconfidence despite noise, but through systematic evidence refinement that maintained diagnostic accuracy. This validates that KnowGuard does not indiscriminately trust retrieved knowledge; rather, it employs coherence-aware filtering to prevent noisy evidence from derailing clinical reasoning, ensuring that only contextually relevant information influences the final decision.

**Case study 3: misleading evidence** This case illustrates KnowGuard's ability to overcome initially misleading evidence through systematic information gathering. Round 1 retrieved 6 triplets heavily biased toward geriatric fall risk factors (musculoskeletal disease, low blood pressure, visual impairment—conditions rare in 4-year-olds), achieving only 0.5915-0.6916 priority scores despite high embedding similarity (0.48-0.59). These age-inappropriate priors misled the initial reasoning, correctly yielding very low confidence (1.0). Crucially, the system did not prematurely commit to these misleading signals. Instead, through 5 rounds of targeted questioning ("loss of consciousness?", "neurological history?", "coordination difficulties?"), the knowledge graph underwent progressive recontextualization: Evidence 3 ("Progressive neurological impairment...delay in achieving developmental milestones") jumped from priority 0.5258 (Round 2) to 0.5355 (Round 3) with LLM score increasing from 0.8 to 0.9, while geriatric-focused Evidence 1 saw its LLM score rise from 0.3 to 0.7 only after reframing "musculoskeletal disease" as pediatric neuromuscular disorder. The breakthrough occurred when Round 5's patient response ("uses hands against knees to stand") triggered retrieval of "Additional neurological manifestations...weakness" (priority 0.5109), enabling recog-

nition of Gower's sign specific to Duchenne muscular dystrophy—a diagnosis invisible in Round 1's evidence pool. The confidence trajectory (1.0→3.0→3.0→5.0) demonstrates that KnowGuard treats misleading evidence not as fatal flaws but as signals for knowledge gap detection, using abstention as a trigger for iterative evidence replacement rather than accepting initial retrieval at face value. This validates the framework's core hypothesis: multi-round interaction transforms misleading priors into diagnostic precision through dynamic evidence reranking.

## I   CLINICAL VALIDATION STUDY

To evaluate the clinical appropriateness and safety of our system's diagnostic decisions under imperfect knowledge graph conditions, we conducted a structured validation study with practicing physicians. This validation focuses on two critical dimensions: (1) the appropriateness of abstention timing and diagnostic confidence levels, and (2) the effectiveness of evidence utilization despite knowledge graph limitations.

### I.1   STUDY DESIGN AND PROTOCOL

#### I.1.1   PARTICIPANTS

We recruited four licensed physicians (Physicians 1-4) from two tertiary hospitals to independently review the three diagnostic cases presented in Section H. All participating physicians have clinical experience ranging from 5 to 15 years, with specialties in internal medicine and emergency medicine. Due to the ongoing review process, specific institutional affiliations are withheld.

#### I.1.2   VALIDATION PROTOCOL

Each physician was presented with the complete diagnostic dialogue for all three cases, including:

- Patient presentation and symptom progression
- Retrieved evidence from the knowledge graph at each interaction round
- System's confidence scores and decision rationale
- Final diagnosis with supporting reasoning

For each case, physicians completed a structured questionnaire assessing:

1. **Evidence Quality**: Whether the retrieved evidence, despite knowledge graph limitations, contributed meaningfully to diagnostic reasoning
2. **Decision Timing**: Appropriateness of the system's decision to provide a diagnosis versus requesting additional information versus abstaining
3. **System Strengths**: Mechanisms by which the system overcame knowledge graph imperfections (evidence filtering, clinical reasoning, iterative refinement)
4. **Overall Performance**: Free-text assessment of the system's clinical reasoning and safety

The questionnaire used a combination of multiple-choice questions (allowing single or multiple selections) and open-ended responses. Questions were designed to elicit specific evaluations of the system's handling of three distinct knowledge graph challenges: incompleteness (Case 1), noise (Case 2), and misleading evidence (Case 3).

### I.2   RESULTS AND CLINICAL ASSESSMENT

#### I.2.1   QUANTITATIVE ANALYSIS

Table 19 summarizes the physicians' responses across the three cases. We report the percentage of physicians selecting each option for key evaluation dimensions.

**Case 1 (Incomplete Knowledge Graph):** All physicians (4/4, 100%) acknowledged that the evidence was helpful despite the knowledge graph lacking the explicit "2-year" diagnostic criterion for

Table 19: Physician responses to clinical validation questionnaire (N=4)

| Question | Case | Response Distribution |
|---|---|---|
| *Q1: Evidence Quality Assessment* | | |
| Evidence helpful despite limitations | Case 1 | A (helpful): 75%, B (neutral): 25% |
| | Case 2 | A (helpful): 100% |
| | Case 3 | A (helpful): 100% |
| *Q2: Decision Timing & Confidence* | | |
| Appropriateness of confidence/decision | Case 1 | A (appropriate): 75%, C (abstain): 25% |
| | Case 2 | A (timely): 75%, B (early): 25% |
| | Case 3 | A (appropriate): 100% |
| *Q3: System Strengths (Multiple Selection)* | | |
| Mechanisms for overcoming KG limits | Case 1 | A (filtering): 100%, B (reasoning): 75%, C (knowledge): 50% |
| | Case 2 | A (diagnosis): 25%, B (confidence): 75%, C (filtering): 100% |
| | Case 3 | A (logical): 75%, B (discriminative): 75%, C (persistent): 75%, D (redundant): 25% |
| *Q4/Q5: Comparative Assessment* | | |
| vs. Baseline system | Case 2 | A (safer): 25%, B (both acceptable): 75% |
| | Case 3 | A (better than misdiagnosis): 50%, B (better than abstain): 25%, C (similar to abstain): 25% |

persistent depressive disorder. Three physicians (75%) found the system's high confidence (5.0/5.0) appropriate, attributing this to the question being based on standard clinical knowledge. However, one physician (25%) suggested the system should have been more cautious or requested additional information when explicit supporting evidence was absent. All physicians identified evidence filtering (100%) and clinical reasoning (75%) as key mechanisms enabling correct diagnosis.

**Case 2 (Noisy Knowledge Graph):** The system's noise filtering capability received unanimous recognition (4/4, 100%), with all physicians noting that effective exclusion of 71% irrelevant evidence (e.g., pelvic inflammatory disease, anthrax, COVID-19) was critical. All physicians (100%) rated the decision to ask about cirrhosis history as timely or appropriate, though one physician suggested earlier incorporation of this question. The dynamic evidence updating mechanism was highlighted by 100% of physicians as valuable for improving diagnostic accuracy.

**Case 3 (Misleading Evidence):** All physicians (4/4, 100%) agreed the system successfully overcame misleading evidence through multi-round interaction. The five-round dialogue was considered appropriate or acceptable by all physicians, with 75% noting that each question contributed meaningful diagnostic value. The system's focus on obtaining Gower's sign was recognized by 100% as the critical decision point. However, regarding question quality, 25% of physicians noted some redundancy. Comparative assessment showed that 50% preferred the system's iterative approach over a baseline that might misdiagnose, while 25% each felt it was comparable to direct specialist referral.

### I.2.2 QUALITATIVE INSIGHTS

Physicians provided substantial free-text feedback highlighting both strengths and areas for improvement:

**Strengths Identified:**

- **Evidence Filtering**: "The LLM-based filtering effectively removed noise while retaining clinically relevant evidence" (Physician 4, Case 2)

- **Dynamic Reasoning**: "The system demonstrated strong clinical reasoning by dynamically updating evidence after obtaining the cirrhosis history" (Physician 4, Case 2)

- **Diagnostic Persistence**: "Multi-round interaction effectively addressed diagnostic ambiguity and progressively focused on disease-specific features" (Physician 3, Case 3)
- **Safety Consciousness**: "The system's cautious approach in Case 3, maintaining moderate confidence until obtaining Gower's sign, reflects appropriate clinical conservatism" (Physician 4, Case 3)

**Concerns Raised:**

- **Evidence Transparency**: "While the diagnosis was correct, the reliance on pre-trained knowledge when KG evidence is incomplete may raise questions about reasoning traceability and evidence sources" (Physician 4, Case 1)
- **Differential Diagnosis Completeness**: "Although DMD was correctly diagnosed, the system should acknowledge that recurrent falls in children could involve co-existing conditions (e.g., vision problems, cardiovascular issues, head trauma). The diagnosis appeared too absolute without ruling out differential diagnoses" (Physician 4, Case 3)
- **Initial Question Strategy**: "In Case 2, incorporating additional contextual factors in the initial round (Round 0) could have accelerated the diagnostic process" (Physician 3, Case 2)

### I.2.3 INTER-RATER AGREEMENT

Despite the small sample size, we observed notable consistency in physician assessments:

- **Evidence utility**: 100% agreement (4/4) that knowledge graph evidence remained valuable in Cases 2 and 3 despite imperfections
- **Filtering effectiveness**: 100% agreement (4/4) that noise filtering in Case 2 was successful
- **Diagnostic accuracy**: 100% agreement (4/4) across all three cases that final diagnoses were clinically correct
- **Safety**: No physician raised critical safety concerns about any diagnostic decision

The primary divergence occurred in assessing the appropriateness of confidence levels in Case 1, where 25% preferred more conservative handling when explicit evidence was lacking.

### I.2.4 SUMMARY

This clinical validation study demonstrates that our system's design enables robust diagnostic performance even under imperfect knowledge graph conditions. Physicians consistently recognized three key capabilities: (1) effective evidence filtering to remove noise, (2) strategic use of iterative interaction to gather critical information, and (3) appropriate integration of clinical reasoning when knowledge graphs are incomplete. While the small sample size (N=4) limits generalizability, the unanimous agreement on diagnostic correctness and safety, combined with specific praise for the system's handling of knowledge graph limitations, provides preliminary validation of our approach. The identified areas for improvement—particularly regarding evidence transparency and differential diagnosis articulation—will inform future system refinements.

