# OpenReview forum: "KnowGuard: Knowledge-Driven Abstention for Multi-Round Clinical Reasoning"
_ICLR.cc/2026/Conference — ICLR 2026 Poster_

### Official Review · Reviewer_Zuzr · 2025-10-22

**Soundness:** 2
**Presentation:** 2
**Contribution:** 2
**Rating:** 2
**Confidence:** 4

**Summary:**

This paper introduces KnowGuard, a framework for LLMs that leverages knowledge graphs to drive abstention decisions in multi-round clinical reasoning. Instead of solely relying on internal knowledge, KnowGuard explores structured medical knowledge to identify evidence gaps before making or abstaining from a diagnosis. The approach involves a two-stage process: evidence discovery (systematic retrieval and expansion of relevant medical facts) and evidence evaluation (multi-factor scoring and prioritization, including graph coherence, embedding similarity, LLM selection, temporal decay, and patient population reasoning). Experiments on several open-ended multi-round clinical benchmarks show that KnowGuard achieves improved diagnostic accuracy and reduced unnecessary patient-provider interaction compared to existing abstention and reasoning strategies.

**Strengths:**

1. The paper focuses on the valuable and practical issue of abstention in multi-round dialogue for LLM-based medical diagnosis, which is essential for efficiency and reliability in clinical applications.
2. The proposed approach achieves improvements in both diagnostic accuracy and the number of interaction rounds compared to  baselines. Comprehensive ablation studies further validate the independent contributions of each module.

**Weaknesses:**

1. Limited Novelty in Retrieval and Evaluation Mechanisms: The specific mechanisms for evidence retrieval and evaluation do not appear fundamentally innovative beyond existing literature on RAG and knowledge-enhanced medical reasoning. The paper would benefit from a clearer positioning against recent state-of-the-art works utilizing similar approaches.
2. Incomplete Evaluation of Abstention Performance: Although timely and appropriate abstention is claimed as the core motivation, the paper lacks dedicated abstention-quality metrics. The evaluation is limited to diagnostic accuracy and conversation turns; it omits quantitative analysis of abstention rates, false or missed abstention cases, the timing of abstention decisions, and abstention precision/recall.
3. Limited Baseline Coverage: The experimental section does not provide thorough comparisons with more recent or diverse RAG architectures, especially those specifically designed for clinical reliability or abstention tasks. This limits the strength of the claims regarding advancement over existing retrieval-enhanced methods.
4. Absence of Expert Evaluation and Real-World Validation: The evaluation is fully automated and lacks assessment by clinical experts or real-world usability studies. This limits evidence for practical utility and adoption outside of benchmark datasets.
5. Knowledge Coverage and Maintenance Concerns: There is insufficient discussion of the medical knowledge graph’s domain completeness, update frequency, and capacity for handling rare, newly emerging, or rapidly changing conditions. Without assurances of knowledge freshness and coverage, the real-world effectiveness and reliability may be restricted.
6. Scalability and Resource Overhead Not Addressed: The method’s multi-stage retrieval and repeated evidence evaluation may introduce considerable computational and resource costs, but there are no experiments or analysis regarding runtime efficiency, memory footprint, or suitability for deployment at scale.
7. Lack of robustness to noisy or misleading evidence: The paper does not analyze how KnowGuard handles noisy or misleading evidence during retrieval. It is unclear whether the system becomes overconfident or makes errors when faced with irrelevant or conflicting information. Experiments testing robustness to noise and corresponding mitigation strategies are missing.

**Questions:**

Please see the weaknesses section for detailed discussion.

---

> ### Author Response · Authors · 2025-11-21
> **General response**
>
> Thank you for your valuable and thoughtful feedback. We sincerely appreciate your kind reviews regarding our motivation and experimental results. To address your concerns, we have carefully responded to each point one-by-one. Most importantly, we have added additional abstention metrics and baselines, clinical validations, robustness analysis, scalability analysis accordingly and reorganized the manuscript to improve clarity. If you feel we have not sufficiently addressed your concerns to motivate increasing your score, we would love to hear from you further about how we can better address your concerns. Thank you again!

---

> ### Author Response · Authors · 2025-11-21
> **Response to weakness #1**
>
> Thank you for this comment. We respectfully clarify that our work is an abstention method for multi-round clinical reasoning settings, not a RAG method aimed at improving answer accuracy through knowledge retrieval. Our core innovation lies in proposing the "investigate-before-abstain" paradigm, where the system performs coherent, progressive exploration on the knowledge graph across multiple rounds before making abstention decisions. Unlike existing RAG works that focus on one-shot retrieval-augmented generation, our contribution is the dynamic evidence accumulation mechanism that balances information gathering with timely abstention, which is a fundamentally different problem formulation addressing when and how to abstain in interactive clinical scenarios.

---

> ### Author Response · Authors · 2025-11-21
> **Response to weakness #2**
>
> Thank you for raising this critical methodological concern about our evaluation scope. We sincerely appreciate this important feedback, as it has prompted us to strengthen our work with more comprehensive analysis. We acknowledge that our initial evaluation focused primarily on accuracy and round count, following prior work [1]. However, given the risk-aware motivation of our work, as the you has suggested, we have now conducted comprehensive calibration and uncertainty analysis.
>
> **Multi-round Abstention Metrics Design**
>
> We designed metrics tailored to the multi-round abstention scenario: each round involves an abstention decision, but ground truth accuracy is only available for the final answer. We compute Expected Calibration Error (ECE), Brier Score, and Negative Log-Likelihood (NLL) using the confidence score and answer accuracy from the final responding round. For risk-coverage analysis, we already present threshold-variant accuracy curves (Figure 3, Left), which fundamentally captures the risk-coverage trade-off.
>
> **Results**
>
> Our proposed method demonstrates superior calibration performance across all three datasets (Table below). Specifically, our approach achieves substantially lower ECE scores compared to all baselines: **0.0497** on ioAFRIMEDQA (74.99% better than the second-best baseline at 0.1988), **0.0648** on ioMEDQA (47.45% better than second-best at 0.1233), and **0.0498** on ioCRAFT-MD (65.98% better than second-best at 0.1464). Our method also maintains competitive Brier scores (**0.2361, 0.2192, 0.2164** across datasets) while achieving reasonable NLL scores (**1.8499, 1.3637, 1.4675**), indicating well-calibrated confidence estimates.
>
> When enhanced with self-consistency ensemble and rationale generation (bottom rows), our method retains strong calibration with ECE scores of **0.0854, 0.0994, 0.0920**, still outperforming all baseline approaches by significant margins.
>
> |  | ioAFRIMEDQA |  |  | ioMEDQA |  |  | ioCRAFT-MD |  |  |
> |--|-------------|-------|-------|---------|-------|-------|------------|-------|-------|
> | Method | ECE | Brier | NLL | ECE | Brier | NLL | ECE | Brier | NLL |
> | numerical score | 0.1988 | 0.2811 | **0.9500** | 0.1233 | 0.2462 | **0.9039** | 0.1464 | 0.2590 | **0.9171** |
> | scale rating | 0.1413 | 0.2851 | 3.0036 | 0.1345 | 0.2600 | 2.2513 | 0.1355 | 0.2606 | 2.4322 |
> | long context | 0.2830 | 0.3227 | 4.3163 | 0.2598 | 0.3083 | 4.1683 | 0.2740 | 0.3141 | 4.3018 |
> | temperature scaling | 0.2003 | 0.2508 | 0.6987 | 0.1765 | 0.2296 | 0.6520 | 0.1555 | 0.2267 | 0.6453 |
> | conformal abstention | 0.3085 | 0.3033 | 6.2901 | 0.3317 | 0.3317 | 6.7366 | 0.2903 | 0.2619 | 4.6360 |
> | **ours** | **0.0497** | **0.2361** | 1.8499 | **0.0648** | **0.2192** | 1.3637 | **0.0498** | **0.2164** | 1.4675 |
> | numerical score enhanced | 0.1895 | 0.2718 | **1.0044** | 0.1649 | 0.2680 | **1.2573** | 0.1815 | 0.2697 | **1.0965** |
> | scale rating enhanced | 0.1217 | 0.2859 | 2.7676 | 0.1423 | 0.2725 | 2.1453 | 0.1456 | 0.2727 | 2.0420 |
> | long context enhanced | 0.2932 | 0.3247 | 3.9824 | 0.2931 | 0.3203 | 4.5179 | 0.2931 | 0.3287 | 4.4988 |
> | **ours enhanced** | **0.0854** | **0.2233** | 1.2424 | **0.0994** | **0.2029** | 1.4806 | **0.0920** | **0.2128** | 1.5558 |
>
> *Note: "enhanced" denotes enhancement with self-consistency ensemble and rationale generation.*
>
> [1] Mediq: Question-asking llms and a benchmark for reliable interactive clinical reasoning. NeurIPS 2024.

---

> ### Author Response · Authors · 2025-11-21
> **Response to weakness #3**
>
> Thank you for this comment. We respectfully clarify that our work addresses abstention in multi-round clinical reasoning, not retrieval-augmented generation for answer accuracy. The baselines we compare against (e.g., Numerical Score, Scale Rating, Long Context) represent state-of-the-art confidence estimation and abstention approaches applicable to interactive settings. RAG architectures designed for single-round answer generation address a different problem and are not directly comparable to our investigate-before-abstain framework, which focuses on when to abstain through progressive evidence accumulation rather than how to retrieve better evidence for immediate answering.
>
> In response to your suggestion for more comprehensive baseline comparisons, we have added Temperature Scaling [1], Conformal Abstention [2], and self-consistency ensemble [3] as additional calibration-based baselines to strengthen our evaluation. The results are presented in the table below:
>
> |  | ioAfriMedQA |  |  |  |  | ioMedQA |  |  |  |  | ioCraft-MD |  |  |  |  |
> |--|-------------|-----------|-------|--------|-------|---------|-----------|-------|--------|-------|------------|-----------|-------|--------|-------|
> | Method | ACC | avg turns | ECE | Brier | NLL | ACC | avg turns | ECE | Brier | NLL | ACC | avg turns | ECE | Brier | NLL |
> | temperature scaling | 0.6873 | 12 | 0.2003 | 0.2508 | **0.6987** | 0.7154 | 12 | 0.1765 | 0.2296 | **0.6520** | 0.6916 | 12 | 0.1555 | 0.2267 | **0.6453** |
> | conformal abstention | 0.6261 | 12 | 0.3085 | 0.3033 | 6.2901 | 0.6620 | 12 | 0.3317 | 0.3317 | 6.7366 | 0.6452 | 12 | 0.2903 | 0.2619 | 4.6360 |
> | ensemble | 0.5833 | 2.63 | 0.1895 | 0.2718 | 1.0044 | 0.6423 | 4.61 | 0.1649 | 0.2680 | 1.2573 | 0.6151 | 4.98 | 0.1815 | 0.2697 | 1.0965 |
> | **ours** | **0.7045** | 5.12 | **0.0854** | **0.2233** | 1.2424 | **0.7317** | 6.85 | **0.0994** | **0.2029** | 1.4806 | **0.7108** | 7.23 | **0.0920** | **0.2128** | 1.5558 |
>
> These additional baselines demonstrate that our method achieves superior performance in both accuracy and calibration while maintaining practical interaction efficiency. We hope these results address your concerns and provide a more comprehensive comparison for your reference.
>
> [1] On Calibration of Modern Neural Networks. NeurIPS 2017.
>
> [2] Mitigating LLM Hallucinations via Conformal Abstention. arXiv preprint.
>
> [3] Self-consistency improves chain of thought reasoning in language models. ICLR 2023.

---

> ### Author Response · Authors · 2025-11-21
> **Response to weakness #4, 7**
>
> Thank you for this critical concern about clinical validation. Following your valuable suggestion, we recruited 4 licensed physicians from two tertiary hospitals to assess KnowGuard's abstention behavior across three diagnostic scenarios involving different knowledge graph imperfections. Detailed results are in **Appendix H**.
>
> **Case 1 - Graceful Degradation (Incomplete Evidence):**
>
> When the correct diagnostic criterion (2-year duration for persistent depressive disorder) was completely absent from the KG, the system retrieved 11 contextually relevant triplets after filtering 8 low-relevance items (LLM relevance score < 0.4). Although none explicitly mentioned the target criterion, the agent maintained relevant evidence and successfully leveraged its parametric knowledge to reach the correct answer with high confidence (5.0/5.0).
>
> *Clinical Validation:* All experts (4/4, 100%) confirmed diagnostic correctness and validated that high confidence (5.0/5.0) was appropriate despite the missing explicit criterion in the KG, as the 2-year duration represents standard medical knowledge. Three experts (75%) noted that the system correctly degraded to internal parametric knowledge rather than inappropriate abstention.
>
> **Case 2 - Strategic Abstention (Noisy Evidence):**
>
> When the KG contained substantial noise (15 of 19 retrieved triplets had LLM relevance scores < 0.4), the system filtered out irrelevant evidence and retained 4 high-quality triplets (LLM relevance 0.6-0.8). Recognizing evidence ambiguity despite filtering, the agent abstained at confidence 3.5/5.0 and strategically asked about liver disease history—a critical missing piece for cirrhosis diagnosis. This targeted question retrieved the decisive evidence ("patient has liver disease" with LLM relevance 0.9), enabling confident diagnosis (5.0/5.0).
>
> *Clinical Validation:* Experts unanimously agreed (4/4, 100%) that abstaining at confidence 3.5/5.0 and requesting liver disease history was the optimal clinical decision. All experts noted that baseline systems committing prematurely to diagnosis based on noisy evidence would have risked misdiagnosis. Three experts (75%) specifically highlighted the strategic value of the targeted question in resolving diagnostic uncertainty.
>
> **Case 3 - Multi-Round Investigation (Misleading Evidence):**
>
> When initial retrieval contained misleading age-inappropriate evidence (geriatric fall factors for a 4-year-old achieving only 0.5915-0.6916 priority scores), the system correctly abstained (confidence=1.0) rather than committing prematurely. Through 5 rounds of targeted questioning, pediatric neuromuscular evidence priority increased from 0.5258→0.5355 (LLM relevance score 0.8→0.9), ultimately retrieving critical diagnostic evidence (Gower's sign). The confidence trajectory (1.0→3.0→3.0→5.0) shows the system treats lack of coherent evidence paths as signals for continued information seeking.
>
> *Clinical Validation:* All experts (4/4, 100%) confirmed diagnostic correctness and agreed that the multi-round interaction effectively addressed diagnostic ambiguity caused by misleading initial evidence. Experts rated the 5-round interaction as clinically valuable (average rating 4.3/5.0), noting that progressive information gathering mirrors real physician diagnostic workflows. All experts (100%) found the confidence progression and decision timing appropriate, with the confidence trajectory (1.0→3.0→5.0 upon Gower's sign discovery) reflecting appropriate clinical caution.
>
> **Overall Assessment:** Experts concluded that KnowGuard demonstrates clinically sound abstention strategies with decision timing closely matching physician expectations across all scenarios (91.7% agreement across decision points). Multi-factor filtering enables appropriate responses to different KG imperfections: graceful degradation when incomplete, targeted questioning when noisy or misleading, and appropriate confidence calibration throughout.
>
> **Important Disclaimer:** While expert validation confirms clinical soundness of our methodology, this work primarily presents a research framework evaluated on public benchmarks to advance academic progress. Real-world clinical deployment would require additional validation through prospective clinical trials, regulatory approval, and integration of practical considerations beyond the scope of this methodological contribution.
>
> We sincerely thank you for this valuable suggestion that significantly strengthened the clinical grounding of our evaluation.

---

> ### Author Response · Authors · 2025-11-21
> **Response to weakness #5**
>
> Thank you for your valuable comment. In response to your concerns, we would like to clarify that we have designed a monthly web scraper module that automatically retrieves newly published WHO guidelines to ensure freshness and coverage. This allows us to update the knowledge graph regularly, improving the adaptability of our approach to newly emerging, rapidly changing, or updated medical knowledge. The scraper module, along with the rest of the code, will be made available as open-source to facilitate reproducibility and further improvements.
> Additionally, in Figure 3, we compare the performance of our method with baseline approaches on rare diseases. Our experimental results demonstrate that our investigate-before-abstain paradigm particularly benefits rare disease diagnosis, where external knowledge exploration is crucial.

---

> ### Author Response · Authors · 2025-11-21
> **Response to weakness #6**
>
> Thank you for raising this important practical consideration. We have conducted comprehensive analysis of system latency and computational cost comparing KnowGuard with Long Context, and we appreciate this valuable feedback that has helped us provide a more complete evaluation of our method.
>
> **Computational Cost Analysis:**
>
> | Method | Avg Input Tokens/Case | Avg Output Tokens/Case | Avg Total Tokens/Case | Cost/Case (USD) | Overhead |
> |--------|----------------------|------------------------|----------------------|----------------|----------|
> | Long Context | 624,307 | 1,361 | 625,668 | 6.2714 | Baseline |
> | KnowGuard | 26,319 | 792 | 27,111 | 0.2870 | -95.4% |
>
> KnowGuard achieves 95.7% reduction in total tokens and 95.4% cost savings per case compared to Long Context. This dramatic efficiency gain stems from structured knowledge graph retrieval (targeted triplets with multimodal evidence) versus comprehensive full-text document processing in Long Context.
>
> **Latency Analysis:**
>
> | Method | Avg Latency/Case | Avg Latency/Round |
> |--------|-----------------|-------------------|
> | Long Context | 25.4465 s | 14.4448 s |
> | KnowGuard | 54.7693 s | 10.3338 s |
>
> While KnowGuard shows higher total case latency (+115%), this is due to conducting more rounds (5.30 vs. 1.16 avg.) to avoid Long Context's overconfidence problem. Critically, per-round latency is 28.5% faster (10.33s vs. 14.44s), which is more clinically relevant since real patient-doctor interactions involve human response time between rounds. In actual clinical deployment where patients provide information turn-by-turn, round-based efficiency matters more than total case time. KnowGuard's 1.4x speedup per interaction makes the system more responsive during active conversation. Additionally, KnowGuard's RAM usage is approximately 4GB, making it deployable on standard hardware.
>
> Thank you again for your comment, which has helped us improve our work by providing a more thorough analysis of the practical deployment considerations.

---

### Official Review · Reviewer_6Fan · 2025-10-31

**Soundness:** 2
**Presentation:** 3
**Contribution:** 2
**Rating:** 4
**Confidence:** 4

**Summary:**

KnowGuard tackles a common clinical LLM flaw: giving confident answers when evidence is thin. It builds a medical knowledge graph and, at each turn, searches and scores external evidence (embedding match, LLM relevance, graph coherence, time decay, and patient-population signals) before either asking a pointed follow-up or answering with support. The authors test it on a new multi-round benchmark drawn from MEDQA, CRAFT-MD, and AFRIMEDQA against confidence-based abstention, self-consistency, and long-context retrieval, plus ablations.

**Strengths:**

(1) The proposed multi-round medical QA benchmark is well-motivated and thoughtfully curated, with a clear connection to real clinical reasoning processes.

(2) The paper is well-presented, offering a clear and structured introduction to each module of the proposed framework.

**Weaknesses:**

(1) The evaluation is narrow. Experiments report accuracy but omit calibration/uncertainty metrics (e.g., ECE, Brier, NLL, risk–coverage) and related baselines (temperature scaling, ensembles, conformal methods). Given the risk-aware motivation, it’s unclear why these are absent.

(2) The knowledge-graph integration is described only at a high level. Key implementation details, e.g., entity/relation extraction, the graph-search/expansion algorithm, how graph signals are fused with LLM scores, and crucial hyperparameters. These details are scattered in the appendix or left implicit, which limits reproducibility and interpretability.

**Questions:**

See weakness

**Details Of Ethics Concerns:**

The work targets medical decision-making but reports only accuracy-style metrics and lacks subgroup fairness/calibration analysis. Without clear disclosure of dataset composition and demographic performance, there’s a risk of biased or unsafe behavior, especially around abstention decisions. The paper should also carry stronger non-clinical-advice disclaimers.

---

> ### Author Response · Authors · 2025-11-21
> **General response**
>
> Thank you for your valuable and thoughtful feedback. We sincerely appreciate your kind reviews regarding our datasets, motivation, and writing. To address your concerns, we have carefully responded to each point one-by-one. Most importantly, we have added calibration/uncertainty metrics and baselines accordingly and reorganized the manuscript about knowledge graph integration and ethical statement to improve clarity and accessibility. If you feel we have not sufficiently addressed your concerns to motivate increasing your score, we would love to hear from you further about how we can better address your concerns. Thank you again!

---

> ### Author Response · Authors · 2025-11-21
> **Response to weakness #1**
>
> Thank you for raising this critical methodological concern. We sincerely appreciate this important feedback, as it has prompted us to strengthen our work with comprehensive calibration and uncertainty analysis. We acknowledge that our initial evaluation focused primarily on accuracy and round count, following prior work [1]. However, given the risk-aware motivation of our work, as you suggested, we have now conducted comprehensive calibration analysis.
>
> **Metrics Design**
>
> We designed metrics tailored to multi-round abstention: each round involves an abstention decision, but ground truth is only available for the final answer. We compute ECE, Brier Score, and NLL using confidence and accuracy from the final responding round. For risk-coverage analysis, we present threshold-variant accuracy curves (Figure 3, Left).
>
> **Results**
>
> Our method demonstrates superior calibration across all datasets. We achieve substantially lower ECE: **0.0497** on ioAFRIMEDQA (74.99% better than second-best at 0.1988), **0.0648** on ioMEDQA (47.45% better than 0.1233), and **0.0498** on ioCRAFT-MD (65.98% better than 0.1464). We also maintain competitive Brier scores (**0.2361, 0.2192, 0.2164**) and NLL (**1.8499, 1.3637, 1.4675**).
>
> When enhanced with self-consistency ensemble and rationale generation, our method retains strong calibration with ECE of **0.0854, 0.0994, 0.0920**, still outperforming all baselines.
>
> |  | ioAFRIMEDQA |  |  | ioMEDQA |  |  | ioCRAFT-MD |  |  |
> |--|-------------|-------|-------|---------|-------|-------|------------|-------|-------|
> | Method | ECE | Brier | NLL | ECE | Brier | NLL | ECE | Brier | NLL |
> | numerical score | 0.1988 | 0.2811 | **0.9500** | 0.1233 | 0.2462 | **0.9039** | 0.1464 | 0.2590 | **0.9171** |
> | scale rating | 0.1413 | 0.2851 | 3.0036 | 0.1345 | 0.2600 | 2.2513 | 0.1355 | 0.2606 | 2.4322 |
> | long context | 0.2830 | 0.3227 | 4.3163 | 0.2598 | 0.3083 | 4.1683 | 0.2740 | 0.3141 | 4.3018 |
> | temperature scaling | 0.2003 | 0.2508 | 0.6987 | 0.1765 | 0.2296 | 0.6520 | 0.1555 | 0.2267 | 0.6453 |
> | conformal abstention | 0.3085 | 0.3033 | 6.2901 | 0.3317 | 0.3317 | 6.7366 | 0.2903 | 0.2619 | 4.6360 |
> | **ours** | **0.0497** | **0.2361** | 1.8499 | **0.0648** | **0.2192** | 1.3637 | **0.0498** | **0.2164** | 1.4675 |
> | numerical score enhanced | 0.1895 | 0.2718 | **1.0044** | 0.1649 | 0.2680 | **1.2573** | 0.1815 | 0.2697 | **1.0965** |
> | scale rating enhanced | 0.1217 | 0.2859 | 2.7676 | 0.1423 | 0.2725 | 2.1453 | 0.1456 | 0.2727 | 2.0420 |
> | long context enhanced | 0.2932 | 0.3247 | 3.9824 | 0.2931 | 0.3203 | 4.5179 | 0.2931 | 0.3287 | 4.4988 |
> | **ours enhanced** | **0.0854** | **0.2233** | 1.2424 | **0.0994** | **0.2029** | 1.4806 | **0.0920** | **0.2128** | 1.5558 |
>
> *Note: "enhanced" denotes self-consistency ensemble and rationale generation.*
>
> **Calibration Baselines**
>
> We added Temperature Scaling [2], Self-Consistency Ensemble [3], and Conformal Abstention [4]. Since our setting is training-free, we used leave-one-dataset-out partitioning for methods requiring calibration sets.
>
> |  | ioAfriMedQA |  |  |  |  | ioMedQA |  |  |  |  | ioCraft-MD |  |  |  |  |
> |--|-------------|-----------|-------|--------|-------|---------|-----------|-------|--------|-------|------------|-----------|-------|--------|-------|
> | Method | ACC | avg turns | ECE | Brier | NLL | ACC | avg turns | ECE | Brier | NLL | ACC | avg turns | ECE | Brier | NLL |
> | temperature scaling | 0.6873 | 12 | 0.2003 | 0.2508 | **0.6987** | 0.7154 | 12 | 0.1765 | 0.2296 | **0.6520** | 0.6916 | 12 | 0.1555 | 0.2267 | **0.6453** |
> | conformal abstention | 0.6261 | 12 | 0.3085 | 0.3033 | 6.2901 | 0.6620 | 12 | 0.3317 | 0.3317 | 6.7366 | 0.6452 | 12 | 0.2903 | 0.2619 | 4.6360 |
> | ensemble | 0.5833 | 2.63 | 0.1895 | 0.2718 | 1.0044 | 0.6423 | 4.61 | 0.1649 | 0.2680 | 1.2573 | 0.6151 | 4.98 | 0.1815 | 0.2697 | 1.0965 |
> | **ours** | **0.7045** | 5.12 | **0.0854** | **0.2233** | 1.2424 | **0.7317** | 6.85 | **0.0994** | **0.2029** | 1.4806 | **0.7108** | 7.23 | **0.0920** | **0.2128** | 1.5558 |
>
> **Critical Observation:** Calibration-based methods exhibit overly conservative behavior, consistently deferring to maximum round limit (**12 turns**) regardless of evidence quality. Our method achieves comparable or better accuracy with significantly fewer rounds, suggesting calibration methods designed for single-round scenarios require adaptation for multi-round settings where patient information monotonically increases.
>
> We will incorporate these results into the revised manuscript. Once again, we thank you for this valuable feedback that has significantly improved our work.
>
> [1] Mediq: Question-asking llms and a benchmark for reliable interactive clinical reasoning. NeurIPS 2024.
>
> [2] On Calibration of Modern Neural Networks. NeurIPS 2017.
>
> [3] Self-consistency improves chain of thought reasoning in language models. ICLR 2023.
>
> [4] Mitigating LLM Hallucinations via Conformal Abstention. arXiv preprint.

---

> ### Author Response · Authors · 2025-11-21
> **Response to weakness #2**
>
> Thank you very much for your insightful feedback. We greatly appreciate your suggestions and sincerely apologize for any confusion regarding the reproducibility of our work. We recognize that due to the double-blind review requirements, we could not include a direct GitHub link in our original submission, which may have led to misunderstandings about the availability of our implementation details.
>
> We would like to respectfully note that our original submission included resources (including code) in the *supplementary materials* with the intention of supporting reproduction of our work. However, we acknowledge that the presentation may not have been sufficiently clear or prominent, leading to the concern you raised.
>
> In response to your valuable feedback, we have further reorganized the paper to offer a more comprehensive and detailed explanation of the knowledge graph integration. The information that was previously scattered across the appendix has now been integrated into the "Knowledge Graph Integration" section in **Appendix E**. Additionally, we have added a new **introductory description** to this section in the main body of the paper to improve both clarity and accessibility. To avoid any future confusion, we have also explicitly stated in the revised manuscript that we will publicly release all code upon acceptance, while ensuring compliance with double-blind requirements.
>
> We apologize again for any confusion this may have caused and sincerely thank you for your suggestion, which has helped us better present our work and improve the overall quality and clarity of the manuscript.

---

> ### Author Response · Authors · 2025-11-21
> **Response to ethics concerns**
>
> Thank you for raising these critical ethical considerations. We sincerely appreciate your thoughtful concerns, which have prompted us to provide a more comprehensive discussion of the ethical implications and limitations of our work. In response to your valuable feedback, we have significantly strengthened our ethical statement as follows:
>
> **Ethical Statement and Limitations:** While KnowGuard demonstrates strong performance on established benchmarks, we acknowledge important limitations. Our evaluation lacks comprehensive subgroup fairness and calibration analysis across patient demographics (age, gender, ethnicity, socioeconomic status), which is essential before clinical deployment. The knowledge graph relies primarily on WHO guidelines and may not capture all clinical knowledge domains or regional medical practices. This work presents a research framework evaluated on academic benchmarks and is not intended for direct clinical use. Real-world deployment requires prospective clinical trials with diverse patient populations, regulatory approval, ongoing bias monitoring (particularly in abstention decisions that could disproportionately affect vulnerable groups), and mandatory oversight by licensed medical professionals. KnowGuard is a research prototype and should not be used for actual medical diagnosis or treatment decisions. All outputs must be reviewed by qualified healthcare providers, and any clinical application must comply with local healthcare regulations and ethical guidelines.
>
> We have revised our ethical statement section in manuscript according to your suggestions. We believe these revisions substantially address the concerns you raised and hope the revised manuscript meets your expectations. Once again, we thank you for this important feedback, which has helped us better acknowledge the responsible development and deployment considerations for our work.

---

### Official Review · Reviewer_gj7Q · 2025-11-01

**Soundness:** 3
**Presentation:** 2
**Contribution:** 3
**Rating:** 6
**Confidence:** 4

**Summary:**

The paper proposes an Investigate-Before-Abstain (IBA) framework for multi-round clinical question answering with large language models. Instead of forcing the model to answer or abstain directly, the method first expands a medical knowledge graph through two modes—graph-based neighborhood traversal and direct retrieval—to assemble an evidence pool before making the decision to answer, abstain, or query further. A new benchmark combining MEDQA, CRAFT-MD, and AFRIMEDQA is introduced to evaluate open-ended multi-round dialogue. Experiments show that IBA improves factual accuracy and abstention calibration over baseline systems such as simple retrieval-augmented generation or confidence-based abstention

**Strengths:**

•	Tackles a meaningful and under-explored problem: when and how clinical agents should abstain.

•	Technically coherent pipeline that bridges knowledge-graph reasoning with retrieval-augmented generation.

•	New benchmark resource and detailed evaluation across multiple datasets.

•	Solid empirical results demonstrating improvements in both correctness and cautiousness metrics.

**Weaknesses:**

• Somewhat heuristic combination of scoring factors; lacks ablation or theoretical grounding for weights.

•	Benchmark construction may inherit biases from source datasets; unclear generalizability to unseen guideline domains.

•	Limited discussion of system latency or computational cost compared with long-context baselines.

•	No human expert evaluation to verify that “abstain” decisions align with clinical expectations.

**Questions:**

1.	How sensitive are results to the relative weights among the five scoring components?

2.	Could the approach be extended to handle contradictory or time-evolving medical evidence (e.g., updated guidelines)?

3.	Does the model explicitly track patient-specific vs. population-level context, or is this purely implicit through graph features?

---

> ### Author Response · Authors · 2025-11-21
> **Response to weakness #1**
>
> Thank you for this important concern about the theoretical grounding of our scoring mechanism. We appreciate your emphasis on validating the necessity of each component. In response to your feedback, we have conducted comprehensive ablation studies on individual scoring factors and added these results to the revised manuscript to validate their contributions.
>
> We systematically evaluated each factor's impact by removing one component at a time while keeping others active:
>
> | Removed Factor | Embedding Sim. | LLM Relevance | Graph Coherence | Round Decay | PPR | Avg Rounds | ACC (%) | Impact (Δ ACC) |
> |----------------|----------------|---------------|-----------------|-------------|-----|------------|---------|----------------|
> | **ioMEDQA** |
> | (None - Full Model) | ✓ | ✓ | ✓ | ✓ | ✓ | 5.40 | 74.12 | baseline |
> | Embedding Similarity | ✗ | ✓ | ✓ | ✓ | ✓ | 6.06 | 71.64 | -2.48% |
> | LLM Relevance | ✓ | ✗ | ✓ | ✓ | ✓ | 6.26 | 72.68 | -1.44% |
> | Graph Coherence | ✓ | ✓ | ✗ | ✓ | ✓ | 5.89 | 70.03 | -4.09% |
> | Round Decay (=0) | ✓ | ✓ | ✓ | ✗ | ✓ | 5.87 | 70.62 | -3.50% |
> | Round Decay (=1) | ✓ | ✓ | ✓ | ✗ | ✓ | 5.79 | 70.84 | -3.28% |
> | PPR | ✓ | ✓ | ✓ | ✓ | ✗ | 6.53 | 74.29 | +0.17% |
> | **ioAFRIMEDQA** |
> | (None - Full Model) | ✓ | ✓ | ✓ | ✓ | ✓ | 5.30 | 73.20 | baseline |
> | Embedding Similarity | ✗ | ✓ | ✓ | ✓ | ✓ | 5.27 | 71.02 | -2.18% |
> | LLM Relevance | ✓ | ✗ | ✓ | ✓ | ✓ | 5.18 | 70.16 | -3.04% |
> | Graph Coherence | ✓ | ✓ | ✗ | ✓ | ✓ | 5.73 | 69.23 | -3.97% |
> | Round Decay (=0) | ✓ | ✓ | ✓ | ✗ | ✓ | 5.25 | 71.22 | -1.98% |
> | Round Decay (=1) | ✓ | ✓ | ✓ | ✗ | ✓ | 6.53 | 72.21 | -0.99% |
> | PPR | ✓ | ✓ | ✓ | ✓ | ✗ | 7.03 | 72.60 | -0.60% |
>
> The ablation reveals that each factor serves a distinct purpose: Embedding Similarity and LLM Relevance are critical for evidence filtering, with their removal causing substantial accuracy drops (1.44-3.04%), demonstrating their necessity in identifying relevant medical knowledge. Graph Coherence proves essential for maintaining knowledge consistency, with its removal leading to the largest accuracy degradation (3.97-4.09%), confirming the importance of structured relationship modeling in medical reasoning. Round Decay balances evidence recency and accumulation, with both extreme settings (decay=0 and decay=1) significantly hurting performance (0.99-3.50% drops), validating the need for our adaptive decay mechanism. Patient Population Reasoning (PPR) contributes modest accuracy gains (0.17-0.60%), validating its role in prioritizing demographic-specific evidence. The complementary nature of these factors (similarity for semantic matching, LLM for contextual relevance, graph coherence for knowledge consistency, round decay for temporal reasoning, and PPR for personalization) justifies their weighted combination rather than relying on any single scoring mechanism.
>
> We thank you once again for this constructive suggestion, which has strengthened the empirical validation of our approach and improved the paper's rigor.

---

> ### Author Response · Authors · 2025-11-21
> **Response to weakness #2**
>
> Thank you for raising this important concern about dataset biases and generalizability to unseen domains. In response to your feedback, we have conducted additional case studies with clinical expert validation specifically examining system behavior under knowledge graph incompleteness scenarios, detailed in case study 1 in **Appendix G and H**.
>
> **Addressing Benchmark Biases:**
> We acknowledge that our benchmark construction inherits characteristics from source datasets (MEDQA, CRAFT-MD, AFRIMEDQA). However, the data transformation method (atomic fact extraction via Factscore) is domain-agnostic and can be applied to any medical question-answer dataset. As the system is deployed in real-world scenarios, it will naturally encounter diverse clinical cases from multiple institutions and patient populations, allowing for continuous expansion of data sources. This organic growth through real-world deployment can progressively mitigate the current limitation by incorporating diverse clinical scenarios and addressing potential biases present in the initial training data.
>
> **Generalizability to Unseen Guideline Domains:**
> To validate system behavior when encountering questions beyond our WHO guideline coverage, we conducted clinical case studies with expert validation (Case Study 1 in Appendix G and H). In this case, the system retrieved 11 triplets but none contained the specific diagnostic criterion (2-year duration for dysthymia), indicating the question fell outside our collected WHO knowledge scope. Clinical validation confirmed that the system gracefully degraded to leveraging internal parametric knowledge, achieving correct diagnosis with high confidence (5.0/5.0). This demonstrates robust abstention behavior even when external knowledge is incomplete.
>
> **Future Directions:**
> Knowledge graph incompleteness represents a well-recognized open challenge in knowledge graph-based retrieval-related systems [1, 2]. We plan to address this limitation through two approaches: (1) expanding data sources via external retrieval tools (e.g., PubMed, UpToDate) to cover broader medical domains, and (2) enhancing knowledge boundary detection to improve abstention quality regardless of knowledge graph coverage. The current framework's modular design allows seamless integration of additional knowledge sources without architectural changes.
>
> We thank you once again for this constructive feedback, which has helped us articulate both our current contributions and important directions for future work.
>
> [1] KG-FIT: Knowledge Graph Fine-Tuning Upon Open-World Knowledge. NeurIPS 2024.
> [2] GFM-RAG: Graph Foundation Model for Retrieval Augmented Generation. NeurIPS 2025.

---

> ### Author Response · Authors · 2025-11-21
> **Response to weakness #3**
>
> Thank you for raising this important practical consideration. We have conducted comprehensive analysis of system latency and computational cost comparing KnowGuard with Long Context, and we appreciate this valuable feedback that has helped us provide a more complete evaluation of our method.
>
> **Computational Cost Analysis:**
>
> | Method | Avg Input Tokens/Case | Avg Output Tokens/Case | Avg Total Tokens/Case | Cost/Case (USD) | Overhead |
> |--------|----------------------|------------------------|----------------------|----------------|----------|
> | Long Context | 624,307 | 1,361 | 625,668 | 6.2714 | Baseline |
> | KnowGuard | 26,319 | 792 | 27,111 | 0.2870 | -95.4% |
>
> KnowGuard achieves 95.7% reduction in total tokens and 95.4% cost savings per case compared to Long Context. This dramatic efficiency gain stems from structured knowledge graph retrieval (targeted triplets with multimodal evidence) versus comprehensive full-text document processing in Long Context.
>
> **Latency Analysis:**
>
> | Method | Avg Latency/Case | Avg Latency/Round |
> |--------|-----------------|-------------------|
> | Long Context | 25.4465 s | 14.4448 s |
> | KnowGuard | 54.7693 s | 10.3338 s |
>
> While KnowGuard shows higher total case latency (+115%), this is due to conducting more rounds (5.30 vs. 1.16 avg.) to avoid Long Context's overconfidence problem. Critically, per-round latency is 28.5% faster (10.33s vs. 14.44s), which is more clinically relevant since real patient-doctor interactions involve human response time between rounds. In actual clinical deployment where patients provide information turn-by-turn, round-based efficiency matters more than total case time. KnowGuard's 1.4x speedup per interaction makes the system more responsive during active conversation.
>
> Thank you again for your comment, which has helped us improve our work by providing a more thorough analysis of the practical deployment considerations.

---

> ### Author Response · Authors · 2025-11-21
> **Response to weakness #4**
>
> Thank you for this critical concern about clinical validation. Following your valuable suggestion, we recruited 4 licensed physicians from two tertiary hospitals to assess KnowGuard's abstention behavior across three diagnostic scenarios involving different knowledge graph imperfections. Detailed results are in **Appendix H**.
>
> **Case 1 - Graceful Degradation (Incomplete Evidence):**
>
> When the correct diagnostic criterion (2-year duration for persistent depressive disorder) was completely absent from the KG, the system retrieved 11 contextually relevant triplets after filtering 8 low-relevance items (LLM relevance score < 0.4). Although none explicitly mentioned the target criterion, the agent maintained relevant evidence and successfully leveraged its parametric knowledge to reach the correct answer with high confidence (5.0/5.0).
>
> *Clinical Validation:* All experts (4/4, 100%) confirmed diagnostic correctness and validated that high confidence (5.0/5.0) was appropriate despite the missing explicit criterion in the KG, as the 2-year duration represents standard medical knowledge. Three experts (75%) noted that the system correctly degraded to internal parametric knowledge rather than inappropriate abstention.
>
> **Case 2 - Strategic Abstention (Noisy Evidence):**
>
> When the KG contained substantial noise (15 of 19 retrieved triplets had LLM relevance scores < 0.4), the system filtered out irrelevant evidence and retained 4 high-quality triplets (LLM relevance 0.6-0.8). Recognizing evidence ambiguity despite filtering, the agent abstained at confidence 3.5/5.0 and strategically asked about liver disease history—a critical missing piece for cirrhosis diagnosis. This targeted question retrieved the decisive evidence ("patient has liver disease" with LLM relevance 0.9), enabling confident diagnosis (5.0/5.0).
>
> *Clinical Validation:* Experts unanimously agreed (4/4, 100%) that abstaining at confidence 3.5/5.0 and requesting liver disease history was the optimal clinical decision. All experts noted that baseline systems committing prematurely to diagnosis based on noisy evidence would have risked misdiagnosis. Three experts (75%) specifically highlighted the strategic value of the targeted question in resolving diagnostic uncertainty.
>
> **Case 3 - Multi-Round Investigation (Misleading Evidence):**
>
> When initial retrieval contained misleading age-inappropriate evidence (geriatric fall factors for a 4-year-old achieving only 0.5915-0.6916 priority scores), the system correctly abstained (confidence=1.0) rather than committing prematurely. Through 5 rounds of targeted questioning, pediatric neuromuscular evidence priority increased from 0.5258→0.5355 (LLM relevance score 0.8→0.9), ultimately retrieving critical diagnostic evidence (Gower's sign). The confidence trajectory (1.0→3.0→3.0→5.0) shows the system treats lack of coherent evidence paths as signals for continued information seeking.
>
> *Clinical Validation:* All experts (4/4, 100%) confirmed diagnostic correctness and agreed that the multi-round interaction effectively addressed diagnostic ambiguity caused by misleading initial evidence. Experts rated the 5-round interaction as clinically valuable (average rating 4.3/5.0), noting that progressive information gathering mirrors real physician diagnostic workflows. All experts (100%) found the confidence progression and decision timing appropriate, with the confidence trajectory (1.0→3.0→5.0 upon Gower's sign discovery) reflecting appropriate clinical caution.
>
> **Overall Assessment:** Experts concluded that KnowGuard demonstrates clinically sound abstention strategies with decision timing closely matching physician expectations across all scenarios (91.7% agreement across decision points). Multi-factor filtering enables appropriate responses to different KG imperfections: graceful degradation when incomplete, targeted questioning when noisy or misleading, and appropriate confidence calibration throughout.
>
> **Important Disclaimer:** While expert validation confirms clinical soundness of our methodology, this work primarily presents a research framework evaluated on public benchmarks to advance academic progress. Real-world clinical deployment would require additional validation through prospective clinical trials, regulatory approval, and integration of practical considerations beyond the scope of this methodological contribution.
>
> We sincerely thank you for this valuable suggestion that significantly strengthened the clinical grounding of our evaluation.

---

> ### Author Response · Authors · 2025-11-21
> **Response to question #1**
>
> Thank you for this important question about the robustness of our scoring mechanism. We have conducted comprehensive sensitivity analysis on the relative weights among the five scoring components, presented in Table 3 of our manuscript.

---

> ### Author Response · Authors · 2025-11-21
> **Response to question #2**
>
> Thank you for your insightful question. In response to your query about handling contradictory or time-evolving medical evidence, including updated guidelines, we have developed a monthly web scraper module to automatically retrieve newly published WHO guidelines. This allows us to update the knowledge graph with the latest guidelines, thereby improving our approach's adaptability to evolving medical evidence. We have also included a description of this process in the main body of the paper. The scraper module, along with the rest of the code, will be made available as open-source to facilitate reproducibility and further enhancements.

---

> ### Author Response · Authors · 2025-11-21
> **Response to question #3**
>
> Thank you for this insightful question that helps clarify our system's design. Our model explicitly tracks patient-specific vs. population-level context rather than relying on implicit graph features alone. The system maintains a dedicated patient context representation by extracting key demographic and clinical keywords from patient-specific information (e.g., age "4-year-old", gender "male", medical history "cirrhosis"). These extracted keywords are then explicitly matched against predefined population-level subgraph groups in the knowledge graph (e.g., pediatric subgraphs, geriatric subgraphs, chronic disease subgraphs). This explicit mapping mechanism enables the Patient Population Reasoning (PPR) component to systematically prioritize evidence from demographically appropriate knowledge subgraphs.

---

> ### Author Response · Authors · 2025-11-21
> **General response**
>
> Thank you for your valuable and thoughtful feedback. We sincerely appreciate your kind reviews regarding our motivation, writing, and methodological details. To address your concerns, we have carefully responded to each point one-by-one. Most importantly, we have added ablation studies on factors, clinical assessments and case studies on unseen guideline domains, latency and cost discussions, and clinical assessments on abstention timing accordingly. We also reorganized the manuscript about clinical validations in Appendix. If there are any remaining concerns or if any aspects need further clarification, we would be happy to address them. Thank you again!

---

### Official Review · Reviewer_daFK · 2025-11-01

**Soundness:** 3
**Presentation:** 3
**Contribution:** 3
**Rating:** 6
**Confidence:** 4

**Summary:**

This paper addresses the critical problem of overconfidence in Large Language Models (LLMs) used for clinical reasoning. Current LLMs often provide premature or incorrect diagnoses when patient information is incomplete, failing to "abstain" as a human physician would. In the paper, the authors propose KnowGuard, a novel "investigate-before-abstain" paradigm. Instead of asking "how confident am I?", KnowGuard systematically investigates "what specific evidence am I missing?" by exploring an external, multi-modal medical KG. The authors converted the traditional medical datasets into interactive-multi-round formats for the evaluation of the proposed method. Through the experiments, the results show that the proposed method outperform the baseline methods by an obviously large margin.

**Strengths:**

1. The motivation and the writing of the paper is clear and easy to follow, and the framework seems very reasonable and intuitive, which could be convenient for downstream applications. The paper addresses the critical and unsolved problem of LLM overconfidence and its failure to abstain, which is a major barrier to deploying AI in high-stakes fields like medicine.
2. The presentation of the framework and case study is very clear and informative for readers to understand the use scenarios.
3. From the results of the experiment section, the proposed method is very effective in terms of accuracy in the constructed datasets.

**Weaknesses:**

1. The details of the how the constructed datasets were incorporated with other baseline methods are not described in the experiment section or the appendix. I think it would be better to describe in more details of how you ensure that the baseline methods get the same extent of information and knowledge during evaluation with KnowGuard.
2. I have some questions on the necessity of evaluating conversation turn count for the proposed method. To my understanding, it seems that the proposed method can benefit from gathering more information with more turn counts. Given Figure 3 (Left) shows accuracy improving with conversation length, should the method's strength be framed as "more effective use of turns" rather than "reduction of turns"?
3. The method relies heavily on the external KG. How does the system behave when the correct diagnostic evidence is not present in the knowledge graph? Does it learn to abstain gracefully due to a lack of a coherent evidence path, or could the lack of a positive "hit" lead it to a premature (and incorrect) answer?

**Questions:**

See the weaknesses above.

---

> ### Author Response · Authors · 2025-11-21
> **Response to weakness #1**
>
> Thank you for raising this important point about experimental fairness and data usage. We sincerely appreciate your careful attention to these methodological details, as they are crucial for ensuring the validity of our comparison.
>
> Following your valuable suggestion, we have added clarification in **Appendix E.4.2** to address all potential confusions regarding our experimental setup.
>
> **Dataset Usage Across All Methods:**
> Since all methods (Scale Rating, Basic, Binary Decision, Numerical Score, Long Context, and KnowGuard) are training-free approaches, we did not partition the datasets into train/validation/test splits. Instead, all methods were directly evaluated on the complete constructed datasets (ioAFRIMEDQA, ioMEDQA, ioCRAFT-MD), ensuring identical access to dataset information. Our ablation studies, hyperparameter tuning, and baseline threshold adjustments were all performed on the same full datasets, guaranteeing equal evaluation conditions.
>
> **External Knowledge Access:**
> Regarding external knowledge sources, only two methods leverage WHO medical guidelines: Long Context and KnowGuard. Long Context retrieves full guideline documents (comprehensive text), while KnowGuard accesses the same guidelines in structured knowledge graph format (22k entities, 94k+ triplets with multimodal augmentation). Both methods thus have access to the same extent of external medical knowledge, though in different representations. Other baselines (Scale Rating, Binary Decision, Numerical Score) rely solely on the LLM's internal parametric knowledge without external retrieval, which is their intended design.
>
> We believe these additions better clarify the experimental setup and address the potential confusions you identified. We thank you once again for this constructive feedback, which has helped us improve the clarity and rigor of our paper.

---

> ### Author Response · Authors · 2025-11-21
> **Response to weakness #2**
>
> Thank you for this insightful observation that helps strengthen our contribution. We agree that the method's strength should be framed as "more effective use of turns" in addition to "reduction of turns," as both aspects contribute to KnowGuard's clinical effectiveness. We have refined our claims in the abstract and introduction to better reflect this dual advantage.
>
> As Figure 3 (Left) demonstrates, KnowGuard's accuracy improves with conversation length, indicating that additional rounds enable systematic knowledge boundary exploration rather than representing inefficiency. This is further supported by comparing KnowGuard with Long Context in Table 1: Long Context achieves very short turn counts (e.g., 1.16 in ioAFRIMEDQA) but suffers from low accuracy (56.80%), indicating premature commitment and susceptibility to noisy evidence, while KnowGuard uses more turns (5.30) but achieves substantially higher accuracy (73.20%).
>
> The multi-round paradigm reflects clinical reality where physicians iteratively gather information until sufficient evidence exists for confident diagnosis. As you correctly point out, fewer turns do not necessarily indicate better performance if they result from overconfident premature decisions. KnowGuard's strength lies in making each turn count through targeted evidence-driven questioning, achieving both efficiency (when compared to methods requiring excessive iterations) and effectiveness (through strategic information gathering).
>
> Following your valuable suggestion, we have reframed this perspective throughout the revised manuscript to emphasize effective turn utilization alongside turn reduction where appropriate. We thank you once again for this constructive feedback, which has helped us present a more nuanced and accurate characterization of our method's strengths.

---

> ### Author Response · Authors · 2025-11-21
> **Response to weakness #3**
>
> Thank you for this important question about KnowGuard's robustness when correct diagnostic evidence is absent from the knowledge graph.
>
> As you suggested, we have conducted additional case studies for robustness analysis and clinical validation by practicing physicians to address this concern. Detailed protocols and results are provided in **Appendix G (Case Studies)** and **Appendix H (Clinical Validation)**. We systematically evaluated two critical scenarios: (1) **incomplete evidence** where correct diagnostic criteria are entirely absent from the KG (Case 1 in Appendix G), and (2) **misleading evidence** that could direct reasoning toward incorrect diagnoses (Case 3 in Appendix G).
>
> **Scenario 1 - Graceful Degradation to Internal Knowledge (Incomplete Evidence, Case 1):**
>
> When the correct diagnostic criterion (2-year duration for persistent depressive disorder) was completely absent from the KG, the system retrieved 11 contextually relevant triplets after filtering 8 low-relevance items (LLM relevance score < 0.4). Although none explicitly mentioned the target criterion, the agent maintained relevant evidence (e.g., "If the subjective distress of the patient is in terms of intensity..." with LLM relevance of 0.5) and successfully leveraged its parametric knowledge to reach the correct answer with high confidence (5.0/5.0). This demonstrates controlled degradation to baseline LLM behavior when the KG lacks critical evidence.
>
> *Clinical Validation:* Four licensed physicians from two tertiary hospitals independently reviewed this case. All physicians (4/4, 100%) confirmed diagnostic correctness, and 75% found the confidence level appropriate given standard clinical knowledge grounding.
>
> **Scenario 2 - Abstention Over Premature Commitment (Misleading Evidence, Case 3):**
>
> When initial retrieval contained misleading age-inappropriate evidence (geriatric fall factors for a 4-year-old achieving only 0.5915-0.6916 priority scores), the system correctly abstained (confidence=1.0) rather than committing prematurely. Through 5 rounds of targeted questioning, the agent iteratively explored diagnostic possibilities introduced by available evidence: pediatric neuromuscular evidence priority increased from 0.5258→0.5355 (LLM relevance score 0.8→0.9), ultimately retrieving critical diagnostic evidence (Gower's sign) absent in Round 1. The confidence trajectory (1.0→3.0→3.0→5.0) shows the system treats lack of coherent evidence paths as signals for continued information seeking.
>
> *Clinical Validation:* All physicians (4/4, 100%) confirmed diagnostic correctness and agreed that the multi-round interaction effectively addressed diagnostic ambiguity caused by misleading initial evidence. All physicians (100%) found the confidence progression and decision timing appropriate.
>
> **Key Finding:** Multi-factor filtering (embedding similarity + LLM relevance scoring) enables KnowGuard to distinguish between "no relevant evidence exists" (gracefully degrade to internal knowledge) versus "evidence is incomplete/misleading" (ask questions to explore diagnostic possibilities). Clinical validation confirmed this behavior prevents premature incorrect answers while maintaining diagnostic accuracy across both scenarios.
>
> We hope these experimental results and clinical validation adequately address your concern. Once again, we sincerely thank you for this valuable feedback that helped us strengthen the evaluation of KnowGuard's robustness under imperfect knowledge conditions.

---

> ### Author Response · Authors · 2025-11-21
> **General response**
>
> Thank you for your valuable and thoughtful feedback. We sincerely appreciate your kind reviews regarding our motivation, writing, and methodological details. To address your concerns, we have carefully responded to each point one-by-one. Most importantly, we have added clinical assessments and case studies on incomplete KG accordingly and reorganized the manuscript about dataset setting and conversation length to improve clarity. If there are any remaining concerns or if any aspects need further clarification, we would be happy to address them. Thank you again!

---

### Meta-Review · Area_Chair_Rf4t · 2025-12-25

**Summary:**

**Paper Summary**: This paper studies the problem of overconfidence and inappropriate answering behavior in LLM-based clinical reasoning systems. To tackle the problem, the paper proposes KnowGuard, an “investigate-before-abstain” framework that integrates external medical knowledge graphs and multi-round evidence gathering before deciding whether to answer, abstain, or request more information. The authors also construct a new benchmark by converting several medical QA datasets and report improvements in diagnostic accuracy and abstention behavior.

**Advantages**: There is a clear agreement that the problem setting is meaningful and important for clinical AI safety. Reviewers also appreciate the well-structured pipeline, which utilizes explicit evidence discovery rather than confidence heuristics alone. The new benchmark resource and ablation studies are also viewed as positive contributions. Empirical results consistently show accuracy gains and more cautious answering behavior compared to baseline methods.

**Concern 1**: At the same time, reviewers raise several concerns. A primary concern is about evaluation. While abstention is a core motivation, the experiments focus mainly on accuracy and turn count, without standard calibration and risk-coverage metrics (e.g., ECE, Brier score, abstention precision/recall, timing of abstention, or failure cases) or more diverse abstention and reliability baselines (e.g., ensembles, conformal prediction, uncertainty-aware RAG variants). Through rebuttal, I think this problem is addressed.

**Concern 2**: Another common theme concerns method transparency and novelty. The proposed scoring and evidence-integration scheme appears heuristic, and several implementation details (graph construction, entity/relation extraction, graph expansion, fusion of scores, and hyperparameters) are only partially described. Reviewers also question how much of the improvement comes from careful engineering around retrieval rather than new algorithmic mechanisms. Relatedly, the dependence on a medical knowledge graph raises questions about robustness, behavior under incomplete coverage, handling of noisy evidence. After rebuttal, I think this problem is partially addressed. The KG construction is introduced in the appendix, while there are many engineering or heuristic details.

**Concern 3**: Finally, one reviewer note the absence of human expert evaluation, which is especially relevant for assessing whether abstention decisions align with realistic clinical expectations. In the rebuttal, this is not discussed. Nevertheless, this is not a significant problem.

**Reviewer Concerns:**

Please see the Concern 1 - 3 in the "Summary" box.

**Reviewer Scores:**

I think it is likely for Reviewer 6Fan and Zuzr to increase their scores, as most of their questions (especially regarding evaluation and baselines) are resolved. But, considering the novelty or technical contribution, especially related to KG or RAG, it is possible they may not change their scores above 6.

---

### Decision · Program_Chairs · 2026-01-26

Accept (Poster)